# Green Retrofitting Simulation for Sustainable Commercial Buildings in China Using a Proposed Multi-Agent Evolutionary Game

Sheng-Yuan Wang , Kyung-Tae Lee * and Ju-Hyung Kim *

Department of Architectural Engineering, Hanyang University, Seoul 04763, Korea; swonwang@hanyang.ac.kr
* Correspondence: ktlee0422@naver.com (K.-T.L.); kcr97jhk@hanyang.ac.kr (J.-H.K.)

**Abstract:** Green retrofit is regarded as an effective environmental measure to reduce greenhouse gas emissions in high energy-consuming commercial buildings. However, the current retrofitting rate of complex structures is lower than the expected rate. This study proposed a method of stimulating the interaction of multiple agents (government, developers, and occupants) involved in the green renovation of China's commercial buildings. To this end, the evolutionary game theory was applied to determine the multiple interaction mechanism of the behaviors of the agents, after which the key factors affecting the contrasting behavior of developers and occupants were demonstrated, and a sensitivity analysis was performed to distinguish detailed set parameters. The major results observed are as follows: (1) occupants are less sensitive to varied conditions owing to their vulnerable economic scale, meaning that a more friendly policy environment is essential to facilitate their support; (2) government financial support, such as subsidies or compensation costs, can strongly induce more positive behavior in developers to promote green retrofit; and (3) life-cycle awareness of developers should be improved as a reasonable energy-saving performance can act as a key motivating factor to support green renovation. This research provided a comparative perspective to that of a public–private partnership model.

**Keywords:** multi-agent evolutionary game; green retrofit; Chinese policies; cost-benefit management; sustainable development

## 1. Introduction

The increase in greenhouse gases emissions and energy crises has emerged as a major global challenge in the 21st century. Particularly, the building sector has attracted attention as a high-ratio contributor to the globally high total energy use (36%) and carbon dioxide emission (39%), and is regarded as a main optimization target industry. Moreover, various stakeholders have agreed on the importance of optimizing this sector. Currently, the number of commercial buildings in China is increasing significantly owing to increasing urbanization and population [1]. Accordingly, the energy consumption of these buildings is high, and can be regularly optimized to address current environmental problems and achieve energy savings. To address climate change, environmental pollution, current inappropriate energy consumption, and the energy crisis, the upgrade and revitalization of green retrofit promotion have emerged as promising solutions. Zheng et al. [2] reported that the energy consumption of commercial buildings in China can be reduced by 24.3%, which will contribute to the environmental protection target of China by 2030. In the last decade, various levels of China's government have promulgated several interrelated policies to incentivize the green retrofitting industry and stimulate the promotion of professional green practitioners.

Compared to new green building construction, green retrofit projects are more complex, uncertain, and riskier [3]. Although nations globally are attempting to implement green retrofit promotions by introducing policies, regulations, and strategies to enhance the

trend of green building retrofit, the promoted ratio of retrofitted buildings has remained at a low level, indicating the existence of a huge gap from the expected situation. The critical success factor (CSF) has been employed in the project management field since the 1970s to present key areas that are essential for accomplishing the environmental protection targets [4]. In addition, owing to low levels in the green retrofit condition, several scholars have employed CSF analysis methods to analyze the reason for the poor and ineffective development of green retrofitting. For example, Liang et al. [5] proposed a two-model social-network-based method using CSF, and revealed the explicit relationship between successful green retrofit and stakeholders. In addition, studies have reported that external conditions, such as clear vision, criteria, and additional financial and political support, act as the main motivation for improving larger-scaled green retrofitting. Darko et al. [6] enumerated and summarized five main quantitative impact factors that affect green building promotion via a systematic review of previous literature, and found that external drivers, corporate-level drivers, property-level drivers, project-level drivers, and individual-level drivers motivate stakeholders to employ green building retrofitting. Furthermore, using the preferred reported items principle and the meta-analysis method, they found that low investigation frequency is a limitation that should be addressed, and concluded that "Commitment of all project participants", "Mandatory requirements", "Integrated design", "Cooperation between stakeholders", and "Adequate incentives" are the five vital CSFs affecting green building promotion. In addition, the participation of governments and stakeholders significantly encourages larger-scaled green building retrofitting, and the interactions between these two agents are not relatively independent, indicating that governments can increase the desire of stakeholders to devote more efforts to green building retrofitting [7]. Well-developed countries, such as the UK, US, Australia, and China, which have strict and systematic governmental policies and assessment systems, have observed the drivers and CSFs of green building promotion, which has resulted in improved ongoing green building retrofitting or new building trends, such as the Energy Policy Act in USA, Green Deal, Energy Company Obligation and Heat Incentive in UK (International Energy Association, IEA), and the Energy Efficiency in Government Operation Policy Act in Australia. These policies play a significant role as an external motivation to push and lead domestic green building retrofitting.

Developing countries, such as China, have the same green retrofitting potential as Western economies, and the landscape of formulating green building construction in China was promulgated in 1986 with the establishment of the Design Standard for Energy-Saving of Civil Buildings (heating residential buildings), which indicated the beginning of the priotization of green retrofitting by the Chinese government and AEC industry practitioners [8]; however, this policy only focuses on new constructions in the early stage. In 2001, the Technical Specification for Energy Conservation Renovation of Existing Heating Residential Building was announced as the first domestic technical building retrofit criteria. In 2015, the Evaluation Standard for the Green Renovation of Existing Buildings was announced, which mainly focuses on the cost-effectiveness, technical performance, and flexibility of buildings in various climates and zones. Accordingly, since its establishment in 1986, China's green retrofitting policy has developed and evolved continuously (further details in Section 2.2), and various participating agents have devoted varying efforts to its development. However, despite the establishment of considerable regulation and social situation, the current rate of the green retrofitting process in China is comparatively low. Previous research has demonstrated the significance of occupant behavior, of which the character of occupants significantly impacts on the promotion of the green retrofitting of existing buildings. According to a systematic policy retrieval by Liu et al. [9], the Chinese government has attempted to reduce economic incentives since 2010, despite the fact that it is the most effective method to foster green retrofitting. In addition, the encouragement of the establishment of organizations and training of qualified professionals is essential; however, current policies are generally against the market economy, which makes future tendency ambiguous. To address this, Lu et al. [10] proposed an occupant-

oriented retrofit option selection decision-making method, and demonstrated that multiple measures are more profitable than single ones. In addition, they revealed that automatic lighting control, higher temperature point, and lowering plug load can induce a high energy-saving potential, as well as economic profit [10]. Nevertheless, economic analysis has revealed that the value of energy inflation rate and energy price, and tax are dynamic, and cost-effectiveness during static timing may vary with a change in social circumstance. Kim et al. [11], Fan and Hui [12], Yang et al. [13] encouraged the use of a dynamic process based on the evolutionary game method, but failed to categorize details and applications in commercial buildings (detailed comparison in Section 5).

A general classification of commercial buildings refers to the classification of buildings excluding building with industrial, governmental, and residential functions, and focuses on official buildings, educational buildings, hotels, hospitals, and buildings used for other commercial purposes. Compared to residential buildings or new constructions, property management groups manage occupied enterprises as occupants, indicating a contractual relationship between the two agents. Thus, occupants are vulnerable to risk and punishment during the renovation process.

Despite considerable regulations and social state, the current rate of China's green retrofitting process is comparatively low. Previous studies have demonstrated the significance of occupant behaviors, in which the character of occupants extensively affects the promotion of the green retrofitting of existing buildings. Therefore, this study defined a mechanism to address the development of guidelines for government, developers, and occupied enterprises in their positive and negative choices regarding the promotion and development of China's commercial building green retrofitting area.

The study focused on answering the following key questions.

(1) As the government can currently stimulate and manage green retrofitting in a macroscopic way, what is the optimal level for the government to promote green retrofitting and minimize the pressure on the government, including financial and management pressure?
(2) Do punishment value and level affect the promotion process of the objected game? With a change in the punishment level, what is the notable point where a positive effect is transformed to a negative effect? Will excessive punishment be counteractive to the object issue?
(3) Occupants are a comparatively vulnerable group with respect to cost and benefit analysis. Is a large condition impact too sensitive for occupants? What is the optimal strategy that can satisfy the benefit of occupants?

The methodological flow chart of this research is organized as follows (Figure 1). Section 2 reviews previous global and domestic (in China) research. Section 3 illustrates the entire evolutionary game process from assumptions, establishment, equation analysis, and strategy analysis procedure in a realistic and mathematical perspective. Section 4 discusses the comprehensive multiple agent performances and sensitivity analysis result of varying selected parameters in detail and upon a re-adjusted economic environment. Section 5 demonstrates the numerical result and realistic answers in response to pre-established questions and provides considerable policy recommendations.

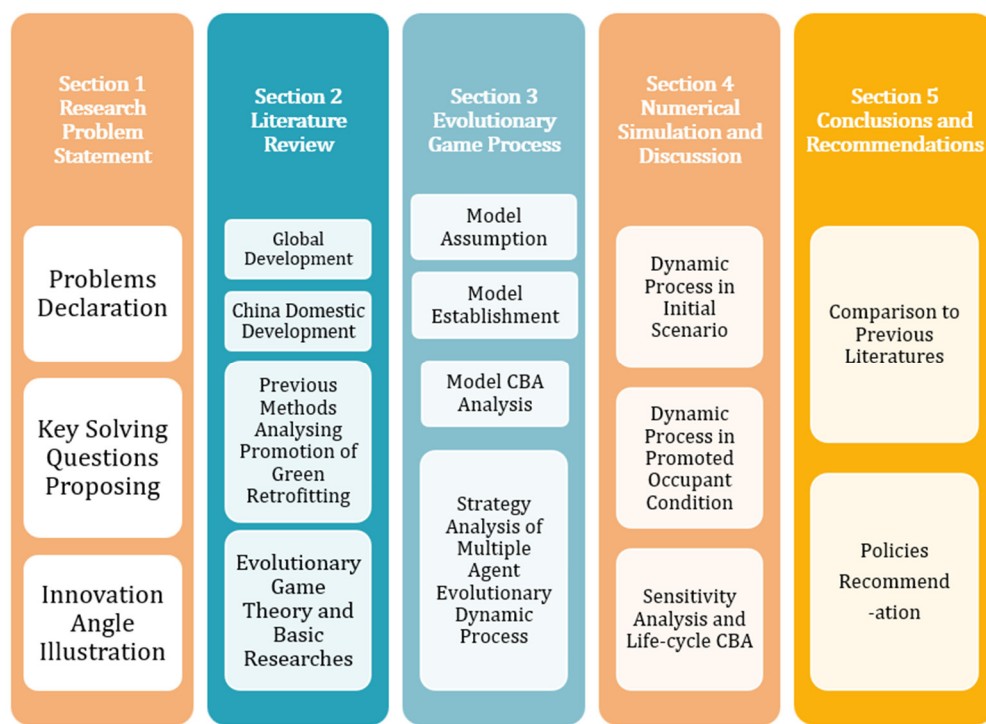

**Figure 1.** Methodological flow chart of this research.

## 2. Literature Review

### 2.1. Global Situation of Green Retrofit Development

Nguyen and Altan [14] reported that there are three ways a green building rating system can stimulate domestic sustainability development to enhance the operational performance of a building: minimizing environmental impact, measuring the effect of buildings on the natural environment, and evaluating and judging the development of buildings objectively. Since the 1970s, the increasing energy and fossil fuel crisis has caused a technically negative impact on the living environment; thus, the significance of sustainable development has gradually attracted considerable attention. The concept of Sustainable Development was first described and defined in the World Commission on Environment and Development (WCED) annual report in 1987, in Tokyo, as "the development that meets the current demand without compromising the requirement of future development". Following the sustainable development trend, green buildings were introduced and referred to gradually, but attracted widespread attention in the 1990s. Building Research Establishment Environmental Assessment Method (BREEAM) was established as the first global green building assessment system in the UK in the 1990s, and it has emerged as a leading assessment system in green building development [15]. In the 2000s, green building rating systems experienced an increase in its adoption. Shan and Hwang [16] reviewed the systematic classification of global green building rating systems, and found that the criteria for the ranking include "energy", "site", "indoor environment", "land and outdoor environment", "material", "water", and "innovation". In 1995, the U.S Green Building Council (USGBC) developed the Leadership in Energy and Environmental Design (LEED), which has emerged as the most well-known and widely employed green building rating system globally. LEED assesses buildings using seven criteria based on 126 available points, and assigns different certification levels: Certified (40–49 points), Silver (50–59 points), Gold (60–79 points), and Platinum (80 points and above) (LEED Reference Guidebook, 2014). In terms of governmental regulations, these aforementioned assessment systems are considered as a powerful tool that can motivate real estate developers and occupants to consider green retrofitting as it offers a landscape criterion for constantly improving green building promotion; however, there

are certain limitations that restrict its further application. These limitations include a lack of globally applicable system, constantly updating an inefficient integrated method, and poor handleability and universality [17,18].

Initially launched in 2004, and re-updated in 2008, 2009, and 2013 into edition V4, LEED standard categorizes green retrofitting promotion as Building Operations and Maintenance (O + M), and it aims to guide the construction of existing buildings, including schools, retail, hospitality, data centers, warehouses, and distribution centers, which are undergoing improvement work or slight construction (LEED, 2018). However, green retrofitting activities can be more complex and complicated compared to new green building construction, and they involve several potential limitations, such as financial barriers, climate change, service changes, human behavior changes, and government policy changes, but there are also equipotent benefits and opportunities [19]. Dalirazar and Sabzi [20] reviewed previous studies using the PESTLE technique, and categorized legal regulation obstacles as "complex procedure", and other obstacles, including complicated contract problems, non-ideal certification, benefit ambiguity, and unstable and precarious regulations. In addition, green building specialists in Sweden, the United States, and New Zealand have reported that social and monetary unwieldiness, including long payback periods, as well as environmental tasks, high initial tasks, and low sustainable building demand, have contributed to negative retrofitting development [20,21].

Although the application of a single policy to uphold green building retrofitting is believed to be difficult [22], nations have consistently proposed initial policy actions. As a highly developed economy, the EU is well known for its high rate of urbanization; thus, it has focused on the green retrofitting of existing building and relatively mature energy efficiency measures [23]. To direct the attention of energy-related sectors towards addressing global warming problems from the early stage, the EU announced the "Construction Products Directive", "Boiler Directive", and "SAVE Directive" in 1989, 1992, and 1993, respectively [24]. By 2000, attention was directed to increasing the thermal insulation performance of existing buildings, and by 2006, reducing energy demand by comparing targeted baseline forecast and mandating low energy consumption threshold were introduced, and attention was directed to renovating 3% of the central government building by utilizing contact; energy auditing; and European Skills, Competences, Qualifications, and Occupations (ESCO) tools in 2011, which represent the main middle-staged action plans and contributed promotions within the EU [24–27]. The energy union strategy (EC, 2015) developed the "energy union package", which highlights the importance of security, energy efficiency, interactions between climate actions and the economy, and innovation research, but this package did not adequately provide renovation guidelines and specific direction. In 2002, a legal policy named the Energy Performance of Buildings Directive (EPBD, 2002/91/EC) was established, and it included a minimum performance criterion for setting a piece foundation and promoting major renovation in the energy sector. The estimation requirement criteria consist of numerous indicators, which are listed in Table 1. In addition, the EU state members developed an energy performance certification scheme to push buyers and tenants into a compulsory perspective 10-year green retrofitting agreement along with a transaction to potentially increase and monitor higher demand of higher green retrofitting renovation force by a threshold of 1000 m$^2$, but this scheme was eliminated in 2010. This new policy posed more economic benefit, while simultaneously ensuring the transparency of the green retrofitting process [28,29]. Nonetheless, the significant difference among European nations resulted in low trust among both green retrofitting developers and occupants despite the decent recast increment, indicating the requirement of a more systematic and comprehensive update of this European rating certificate system [23,30]. In 2010, the EPBD policy was introduced, and it broadened the standard of green retrofitting and included more detailed technical requirements, such as a more enhanced heating, ventilation, and air-conditioning (HVAC) system, and it offers financial funding in association with various financial institutions from the EU. Owing to the significance of the economic burden of green retrofitting and the complexity of existing

buildings compared to new counterparts, the EPBD was updated to include the requirement of at least two additional standards for the green retrofitting scheme, and a more detailed definition toward different building categories was emphasized. In addition, semi-staged packages or the promotion of different levels of green retrofitting represented by energy-efficiency-based measures were re-identified, and the EU governmental regulation was set as a standardized baseline. New zero-energy buildings have been reported as a brilliant concept to synchronously reach the standard GHG emission and reduce economical expenditure for energy demand by achieving a high energy performance and on-site or nearly on-site renewable energy production along with various storage systems since the initial establishment of the concept [31,32]. Similar to other greening schemes, the challenges of objects of existing buildings are larger than those of new ones owing to economic (long payback period), environmental (hard environmental benefit assessment, geography, and climate), technical (lack of information), social (low occupants' recognition and passion), and uncertain barriers [33,34]. Although subsequent policies and long-term retrofitting strategies, such as setting central governmental buildings as a demonstration leader, have been set to foster and enhance cost-effectiveness and push larger-scaled retrofitting promotion, the retrofitting rate in the last five years has remained relatively low owing to the complexity and variation among the Member States and the fact that the cost-effectiveness of the life-cycle needs to be proved [35].

**Table 1.** Indicators of EPBD provisions in 2002.

| Num. | Indicators |
| --- | --- |
| 1 | outdoor and indoor climatic conditions |
| 2 | position and orientation of the building |
| 3 | thermal characteristics of the envelope (including airtightness) |
| 4 | passive solar systems and solar protection |
| 5 | natural ventilation and passive strategies |
| 6 | heating, ventilation, and air-conditioning (HVAC) installations |
| 7 | built-in lighting installations (for the non-residential sector) |
| 8 | own-energy generation |

The U.S. governmental institutions, including federal, state, and local governments, have devoted efforts to implement green retrofitting. To this end, they have attempted to achieve a building energy consumption of 18% and a carbon dioxide emission ratio of 46% in commercial buildings, and these have been stably increased [36,37]. Under the global agreement and plans to reduce GHG emissions and facilitate green process, 50% of the energy demand of current buildings should be reduced, indicating a 75% potential for the green retrofitting of commercial buildings. Nevertheless, the reality of the current financing, sustainable, and information situation, as well as inaccurate policy making, has limited the implementation of promotion on a greater scale [12,38]. Adekanye et al. [39] found a significant synergistic effect between LEED certification and federal policies for the motivation of green retrofitting, and they also classified the main governmental motivation factors as federal policies, USGBC LEED rating system updates, and local policies using a panel data modelling. These policies include recommendations, financial incentives, non-financial incentives, requirements, and density bonuses (the two latter policies have been verified by mathematical data results to promote green retrofitting). Green retrofitting has been promoted using regulation-related efforts, such as the mandatory LEED requirements per square in 2009 California, and financial subsidies in 2013 Nevada. Accordingly, the implementation of the national profile updated from the Energy Policy Act (EPAct) of 2005, the Energy Independence and Security Act (EISA) of 2009, and the American Recovery and Reinvestment Act of 2009 (ARRA) of 2009, has facilitated a reduction in energy consumption and GHG emissions, which is accompanied by an increase in the annual ratio (2%/year to 3%/year) [40]. The action plans of the U.S. government towards achieving green retrofitting have varied over the years. For example, President Obama set the climate protection goal

to increase the energy productivity by over 50% in a 20-year range, and the potential of that policy to provide financial benefit and additional job opportunities was reported [41]. However, from 2017 to 2021, during the four-year tenure of President Trump, the U.S pulled out of the Paris Agreement, and halted its energy-related environmental greening promotion process owing to its traditional energy source development motivation policies, but re-joined the agreement after the power shift to the Biden administration, on claims of promoting infrastructure and achieving a carbon-neutral environment. In the new plan set by the Biden's administration, existing commercial buildings are set as landmarks, and this was initiated by mainly updating the energy sector, including the use of more domestic-made and lucrative and clean HVAC and lighting system to reduce cost for both individual families, and state, local, and city governments [42]. Moreover, new commercial buildings were legislated to meet the new-zero energy standards by 2030 to match the global trend of reducing GHG emission, indicating the high new zero-energy retrofit potential of existing American buildings.

However, the establishment of policies by the government without awareness and initiations by governmental institutions, occupants, and developers can hinder the implementation of regulations and retrofitting plans. For commercial buildings, shareholders and agents of main commercial activities form a triangular relationship with governmental institutions (mainly state government), property management companies (PMCs), and occupants. Regulations, information, and incentives are described as sticks, tambourines, and carrots to define, improve, and influence the promotion of building policies [43,44]. Particularly, a lack of understanding among occupants owing to the huge uncertainty and diversity in the data collection stage results in inaccuracies and difficulties [45]. Previous studies have confirmed that the misled free-ride effect due to the complexity of occupants can cause additional economic and environmental waste beyond the actual meaning of green retrofitting; moreover, the ambiguous benefit distribution among multiple shareholders can also act as obstacles to future revolution [46,47].

*2.2. Green Retrofit Development Situation in China*

Developing countries, such as China, have the same green retrofitting potential as Western economies. The landscape of formulating green building construction in China was promulgated in 1986 with the establishment of the Design Standard for Energy-Saving of Civil Buildings (heating residential buildings), which marked the beginning of the priotization of green retrofitting by the Chinese government and AEC industry practitioners [8]; however, they focused mainly on new constructions in the early stage. In 2001, the technical specification for the energy conservation renovation of existing heating residential building was announced as the first domestic technical building retrofit criteria. In 2015, the evaluation standard for the green renovation of existing buildings was announced, and it mainly focused on achieving cost-effectiveness, technical performance, and flexibility in various climates and zones.

In 2020, virtual actions were initially implemented during the 11th Five-Year Plan (FYP), which is widely referred to as China's national orientation and governmental highlights, and a retrofit of 39,000 communities plan was set. Liu et al. [48] summarized China's green retrofitting policies into six categories, which included direction-based policies (DPs), regulation-based policies (RPs), knowledge and information policies (KIPs), evaluation-based policies (EPs), financial support policies (FPs), and organization and professional training policies (OPPs), among which DPs acts as a guiding background for the national development direction formulated by the Chinese government, and it implements other policies unconditionally. The roles of these policies are highlighted in Table 2. The lack of awareness and low-level popularity of green retrofit are considered as the most fundamental factors that have limited the development of green retrofitting. Hence, among the interactive relations of the six policy categories, KIPs and OPPs potentially and permanently advance and offer the persistent foundation for RPs, FPs, Eps, and OPPs. Thus, for both

the developers and occupants, the benefit is regarded as the core criteria of construction, and FPs offer the most powerful and crucial pattern for other policies.

**Table 2.** Vital policy establishment of China's governmental institutions.

| Effective Time | Categories | Competent Department | | Policy Content |
|---|---|---|---|---|
| 1986 | EP | MOC | ■ | First landmark of the awareness of green construction |
| 2002 | DP | MOC | ■ | 10th Five-Year Plan for building energy conservation regulation |
| 2006 | RBP/OPP | MOC | ■ | Provisions on the administration of energy conservation for civil buildings |
| 2007 | FP | MOF | ■ ■ | Reward funds for the management of heating metering and energy-saving (Interim procedure) Retrofit in Northern central heating district for residential buildings |
| 2008 | RP/FP | SCPRC | ■ | Regulations on energy-saving measures for civil buildings |
| 2008 | RP | MOHURD | ■ | Acceptance of heating metering renovation project for existing residential buildings in the northern heating area |
| 2009 | EP/RP | MOHURD | ■ | Technical specification for the energy conservation renovation of public buildings |
| 2010 | KIP | MOHURD | ■ | Guideline for the promotion of the energy-saving renovation technology of existing buildings |
| 2012 | FP | MOF | ■ ■ | Administration of subsidies for the energy conservation renovation of existing buildings (Interim procedure) Residential buildings in hot summer and cold winter zone |
| 2013 | EP | MOHURD | ■ | Technique guidelines of energy efficiency retrofitting for residential buildings |
| 2015 | EP | MOHURD | ■ | Energy performance evaluation of the heating systems of existing buildings |
| 2016 | EP | MOHURD | ■ | Green retrofitting rating system of existing buildings |
| 2017 | EP | MOHURD | ■ | Energy efficiency retrofitting performance results report guideline publishment |
| 2022 | EP | MOHURD | ■ | Maintenance and retrofit guidebook for existing buildings |
| 2022 | KIP | MOHURD | ■ ■ | The desire to accelerate efforts to tackle key and core technologies and industrial application Plans to establish a life-cycle green construction system |

Among these policies, DPs mainly refer to FYP policies containing the most directive and national plans generated by China's central government every five years, which identifies targets and vital guidelines for the next five years. Wu et al. [49] reviewed previous policies and literature using co-word analysis, and found that the Ministry of Housing and Urban–Rural Development (MOHRUD), which is a rural and core department, accounts for the establishment, realization, and management of 90.45% of the total green retrofitting policies. In addition, MOHRUD has been re-centralized to be responsible for green building policy making after promotion attempts by the cooperation of various government departments, such as MOF, the Ministry of Science and Technology (MOST), the State Economic and Trade Commission (SETC), and other departments. As increasing the energy efficiency performance and GHG emission has been set as the vital mission of MOHURD, measures such as retrofitting existing buildings (2001) and government official buildings (2004) have been employed for continuously updating the green building retrofit assessment system (2009, 2013, and 2022). For example, in most Northern cities, the standard for heating systems with centralized heating systems and complications was set

to over 400 million m$^2$ in 2001. In addition, the 12th FYP is projected to increase the energy efficiency performance of over 60 million m$^2$ of housing. Furthermore, during the 11th FYP, a threshold target of 25 to 10% was set to be achieved from three-level China cities. From the pilot perspective, there are plans to increase the energy efficiency of 100 million m$^2$ public buildings in the 13th FYP, indicating the great ambitions of the Chinese government.

RPs act as a specific and mandatory object for promoting renovation activities of green retrofitting. Since 2001, principles and methods of green retrofitting have been established, and external insulation performance, air quality, building envelope, and window insulation performance have been considered until a recent research study on green retrofit was published in 2022, which contains guidelines on safety performance, building envelope and retaining structure, energy efficiency performance of HVAC system, and the performances of water heating system, and the sustainable energy design sector after successively updated editions.

The building model and the pilot green retrofit project are common KIP promotion measures of the Chinese government. Since the 1980s, extensive attention and efforts have been devoted to pilot projects by the Chinese government, as they are expected to provide effective broadcasting performance. In addition, the significance of the pilot project has been highlighted in several documents, and is expected to increase the public awareness and popularity of green retrofitting, such as the building energy-saving technology policy, the building energy conservation and green building development 13th five-year plan, and opinions of the state council on strengthening ageing in the new era.

EPs comprise the energy efficiency performance testing, auditing, assessment, and targeting of existing buildings. Most newly built buildings are regulated to meet the energy-saving criteria after the final measurement of local administration institutions; however, the government only provides guiding standards and specific thresholds for existing buildings rather than mandatory requirements. Although awareness and emphasis on green building for new construction have been implemented relatively early, and the fact that new buildings have simultaneously met the assessment standards, the official standard for existing building was implemented in 2016, and retrofitted buildings are rated as one-star, two-star, and three-star levels.

Major economic support managed by the local government, e.g., specialized funds and tax subsidies, is distributed according to documental multi-criterions. Financial administration ministries can promote energy efficiency performance and decrease GHG emissions by minimizing tax adjustment [50]. Nevertheless, direct financial support, such as the three types of subsidies in the management of incentive funds for heating metering and energy-saving retrofit of residential buildings in the Northern Heating District, can be used as an interim procedure. In advanced western economies, high initial investment, long payback periods, balancing the risk of private properties, and public–private partnerships (such as energy service companies (ESCO)) are universally implemented to overcome stress and take responsibility of the entire retrofit process, from design to maintenance. FP motivates and stimulates more investors to promote green retrofitting and increase energy efficiency performance; however, the currently low qualified and retrofitted ratio indicates huge improvement potential and an appropriate update of prevailing FPs [51]. Moreover, as professional green retrofit institutions are undertaking incremental responsibilities to promote green retrofit, OPPs facilitate more professional abilities and human resources.

The regulation of the top-down model of China may improve the effectiveness of the state and the local government, which causes potential problems in the enforcement, continuity, consistency, and flexibility of retrofit targets and coordination mechanisms [52]. In addition, the externality of diverse shareholders indicates the significance of policy instruments [53]. Moreover, a tough capital flow situation, such as limitations in financial constraints and inadequate expectation for long payback periods, directly affects green retrofitting [54] and indirectly results in the misunderstanding of the evaluation of objects. Furthermore, recent studies have reported that innovative, profitable, and attractive technologies decelerate the rate of China's retrofitting [55]. In addition, the main obstacles to green retrofitting in China include non-ideal development in regulation, finance, in-

formation, and technologies, which indicates demand to engage in optimal green retrofit human resources, assessment systems, financial support systems, and diverse effective technologies and cooperative development modes.

### 2.3. Previous Methods for Analyzing the Promotion of Green Retrofitting

Green retrofit indicates a high initial investment in the early stage to earn future energy-saving and additional benefits, such as social, sustainable, and green-orientated marketing values, and this indicates that there are undefined benefits and low motivation among developers. Broadly speaking, economic benefit contributes to the core motivation of green retrofitting; thus, ensuring its cost-effectiveness using numerical and quantitative methods is significant. Tremendous efforts have been devoted to evaluate the cost-effectiveness of green retrofitting with a focus on the optimization of rating systems [13,14,16,18,21,56–59], decision-making methods [10,12,60–62], single-technique energy-saving potentials [41,63–69], and policy barrier commendations [39,45,53,55,59,62,70–72] (Figure 2). A static agent focuses on one enterprise itself, but not the entire lateral industry, indicating that it is essential to investigate the promotion level of all enterprises in the developer group or enterprise group. Thus, the promotion process is not a "DO" or "Not Do" type of model, but a gradual growth involving the devotion of new enterprises and the exit of old enterprises. Therefore, a dynamic progression and entire social colony research must be solved. In this section, the authors review related works to compare the contributions and highlight the significance of this research.

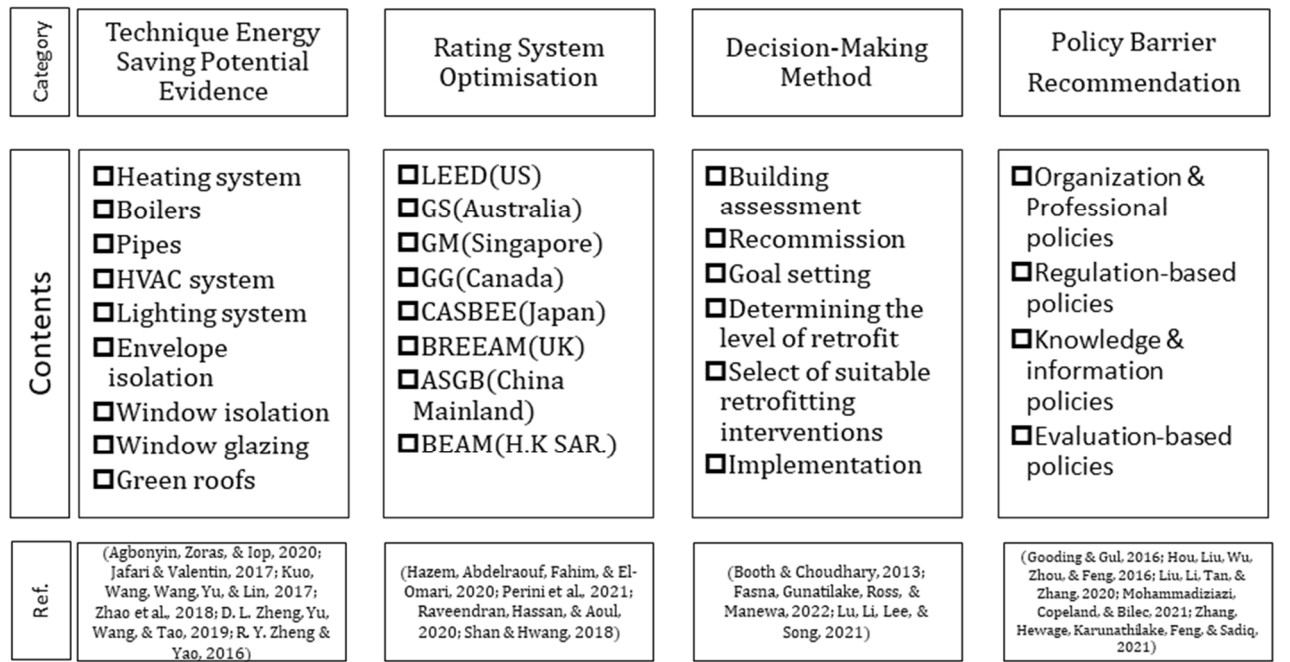

**Figure 2.** Previous research in promoting green retrofitting [1,10,16,48,54,57–59,62–65,68,72–74].

Compared to other building categories, research on commercial buildings is rare owing to the complexity of occupied enterprises and contractual relationships. Property management companies (developers) manage occupied enterprise groups and gain financial benefits from the rent price. The contract details regarding the benefit ratio are complex and vary depending on the scales of occupied enterprise. Generally, compared to the owner or manager of an entire building, developers are more likely to earn more benefits and are responsible for sustaining risks, such as punishment and cancellation compensation. Moreover, after retrofitting construction, maintaining green retrofit habits in individual enterprises is challenging. The real-time procedure for analyzing commercial building objects can be illustrated by implementing an evolutionary game to propose this dynamic process. Cost–benefit analysis (CBA) is typically implemented as a second-stage

tool after technical consideration for economic analysis and to provide optimized decision making. The CBA method is a highly grouped optimized method, which can be achieved by implementing the net present value (*NPV*), the payback period method with discounted cash flow, the annuity method, the dynamic payback period, and the life-cycle cost (LCC) method (applied in Section 3.4 of this research) to systematically compare the economic factors of the green retrofitting plan. Costs include initial investment, operation investment, and maintenance investment.

Technical research has illustrated and verified the efficiency of several energy-saving and profitable options using computer-based simulation software, including EnergyPlus and Autodesk Ecotect(Revit), via the application of actual data. For example, the use of a lighting retrofit window film coating, updating a chiller plant, and installing a building management system have been verified to be effective for saving energy using BIM technology and EnergyPlus software [75]. Individual retrofit measures are widely categorized into positive retrofit measures (PRMs) and negative retrofit measures (NRMs). PRMs involve upgrading the heating system, chillers, boilers, and pipes of HVAC system water piping system and lighting systems, while NRMs involve retrofitting building envelope isolation, windows, and green roofs. Among these techniques, an HVAC system is regarded as the major energy consuming part of entire building facilities, accounting for over 50% of the entire energy consumption [76]. In addition, energy-efficient window glazing and window shading devices have been reported to efficiently promote wall insulation via new negative green design and replacement [77], but a lack of historical data has limited the further application of this measure. In addition, HVAC-based retrofitting measures, such as the use of a more effective HVAC system, a reasonable setting point, and night ventilation, have been reported to be effective for improving green retrofitting levels. Thus, the replacement of ongoing building lighting facilities, particularly incandescent lamps and the compact fluorescent LED lamp, to LED has proved to be environmentally friendly and cost-effective. According to historical data and retrofitted results observed in Chongqing, a major city located in south-western China, among four common Chinese retrofit technique options, the HVAC system requires the lowest investment, and the lighting system exhibited the highest annual economic benefit, followed by the HVAC system, the power supply and distribution system, and the power system, as well as the second shortest payback period [74]. These studies have highlighted the technical feasibility of green retrofit options; however, in an actual retrofitting scenario, developers tend to consider multiple options among diverse options, as more managemental decision making is essential.

Lu et al. [78] developed a creative energy-based decision-making framework using integrated environmental solutions within a virtual environment (IES-VE) and building information modelling technique based on an educational facility as a case study, and reported the relative cost-effectiveness when various retrofit measures were combined. In addition, they found that the combination of a simple scenario combined with multiple retrofit measures, particularly energy-saving measures, was not relatively more effective than individual independent retrofit techniques [60]. The energy-saving performance gap between summed individual measures and simulated scenarios indicates the high beneficial potential of the retrofit of a lighting system and the significance of the climate of the target building object. Shen et al. [79] developed an easy-to-use decision-making method using the agglomerative hierarchical clustering method and dynamic SimBldPy modelling to support the selection of various energy retrofitting measures. Ascione et al. [80] proposed a widely applicable available method for various types of buildings to achieve optimal cost-benefit balance by applying EnergyPlus analysis and MATLAB optimization. Zheng et al. [54] defined a new angle for utilizing the internal energy rate with varying retrofit levels. Based on an actual case study of existing hotel buildings, Fasna et al. [73] developed a decision-making process to assist in the retrofitting process; however, the investment amount requires core consideration. Although this method is feasible for on-site retrofitting, they could not technically determine the best option, and it may ignore life-cycled beneficial options for high investment price.

Research on evolutionary game is rare, but studies have focused on residential buildings whose proportion is higher than that of the public–private partnership (PPP) model, and the modelling of commercial buildings has revealed that the response of enterprises significantly affects the benefit of developers. In the PPP model, more policies are recommended to reduce the investment cost of renovation and to provide more appropriate incentive measures. Studies have reported the synergistic relationship between government and investment groups, as well as the effect of this relationship on the power of incentive measures [13]. However, as members of the government are also shareholders of ESCO enterprises, the benefit of the investment group and government groups need to be further classified.

Green retrofitting techniques not only provide economic energy-saving benefits for developers, but also environmental and social benefits, such as GHG savings and essential increase in rent and real-estate price, particularly for commercial buildings, including official buildings, educational buildings, and shopping malls, owing to the enhanced occupation comfort. Nevertheless, financial-based benefits are considered as the most fundamental and significant indicator. Multiple project benefits have been introduced and identified as an effective means to ensure additional benefit and monetary value using a quantitative cash flow method [21,81], and the analysis of results using this methodological framework indicates that complexity caused by semi-shareholders may result in incorrect economic calculation. Hence, the attained benefit should be considered by a multi-agent rather than a one-side developer or investor. In the case of multiple attending agents for commercial building project operation, governments, developers, occupants, and presently implemented ESCO enterprises have diverse benefits through the single project itself, indicating that the benefit should be considered, not by simply summing up, but by creating separations based on the devoted contribution.

### 2.4. Evolutionary Game for Green Retrofit Problems

Morgenstern and Von Neumann [82] initially proposed the game theory, which has become the landmark of the development of the modern game theory, and has significantly impacted the economic and management field. The game theory is utilized to solve stably increasing economic problems, and has emerged as a fundamental tool in economic analysis. The game theory focuses on the dynamic behaviors of agents within the interactive benefit-shared project, and holds the ability to formulate hypotheses and predictions. Nevertheless, the traditional game theory requires the verification that all assumptions should be fully rational, which only exists in ideal situations. Institutions or stakeholder groups can perform decision-making activities according to the available accepted information, which indicates limited rationality [83]. The evolutionary game theory can optimize the obstacles of the traditional game theory as limitations in policymakers' perception and lingual ability can result in non-ideal rationality. Thus, the evolutionary game theory promotes a dynamic game balance compared to the static balance of traditional methods. The evolutionary game theory is widely used to analyze social and economic developing problems.

As a theoretical foundation of evolutionary game theory, the theory of biological evolution and the theory of genetics were proposed by Darwin and Lamarck, respectively. The application of the evolutionary game theory to actual social problem solving was first performed by Smith [84]. The evolutionary stability strategy and the replication dynamic developed the concept and availability of this method. To date, the evolutionary game theory has developed rapidly based on a relatively complete theory. From the 1980s to the 1990s, under the guidance of Smith's publication "Evolution and Game Theory", the evolutionary game theory attracted increased attention, and the research direction shifted from symmetric games to asymmetric games. Originally used in biology, the evolutionary game theory has also attracted widespread attention in fields, such as political science, sociology, and economics. Cohen et al. [85] developed a new methodological method based on Prisoner's Dilemma model to define the barriers to Israel's energy efficiency promotion activity and provide recommendations for policy-makers. Fan and Hui [12] proposed a quantitative method to verify the effectiveness of new green building promotion between interactions of governmental subsidies and develop-

ers. A previous study illustrated the significance of price premium and the affordability level of incentives. Singh and Mukherjee [86] established a system dynamism simulation method that utilizes the evolutionary method to ensure the interaction result between environmental impact management of regulating groups and stakeholders and provided recommendations for promoting more effective regulation and environment protection performance.

The referred agents used for the calculations in this study are shown in Figures 3 and 4. This research includes an additional occupant interaction illustration to provide double-agent-based evolutionary models.

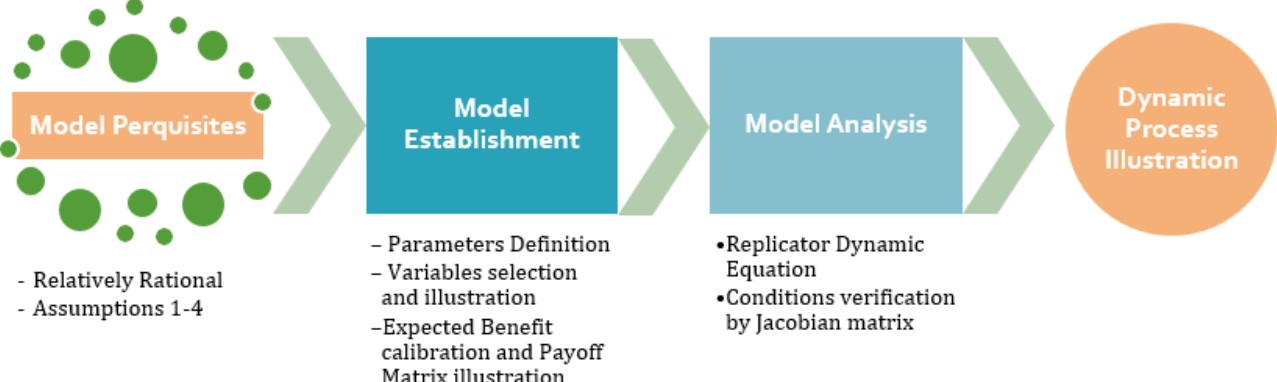

**Figure 3.** Mathematic calculation directions of this research.

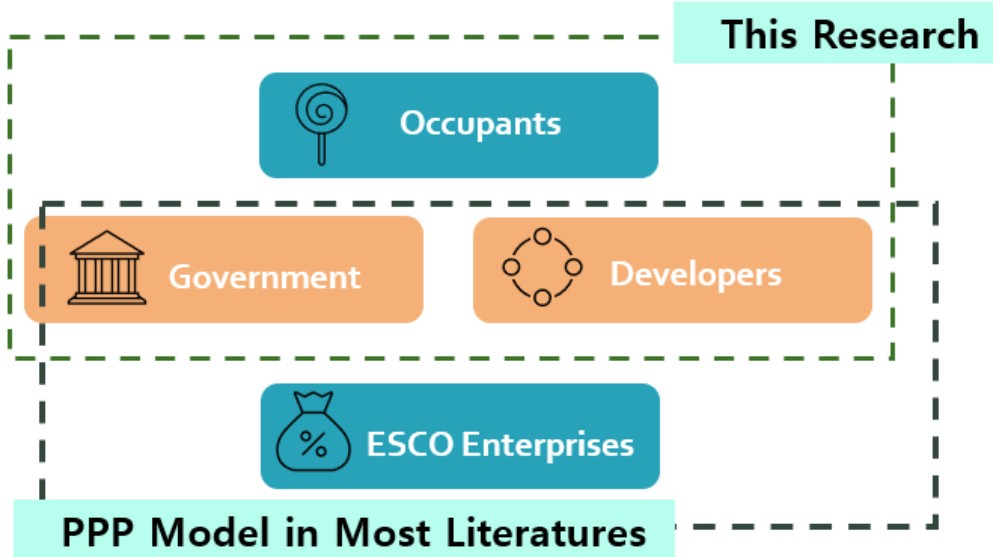

**Figure 4.** Research agents used in most previous literatures.

## 3. Evolutionary Game Model for Promoting Green Building Retrofitting

### 3.1. Model Assumptions

Although the PPP model is highly implemented in China's green retrofitting activity, the green retrofitting potential of enterprises and financial institutions, such as ESCOs and banks, is still enormous. Accordingly, this research aimed to develop a multiple-agent evolutionary game framework that considers governments, developers, and occupant enterprises. Figure 5 shows the interactive relationship among the targeted agents. The basic prerequisite of an evolutionary game is that the participants of the game agents are relatively rational. Based on the evolutionary game theory, the main optional strategies can be adjusted by a considerable scaled group until all game process attains a stable condition, i.e., an evolutionary stable strategy. Consequently, the evolutionary game theory is widely implemented in management territory research.

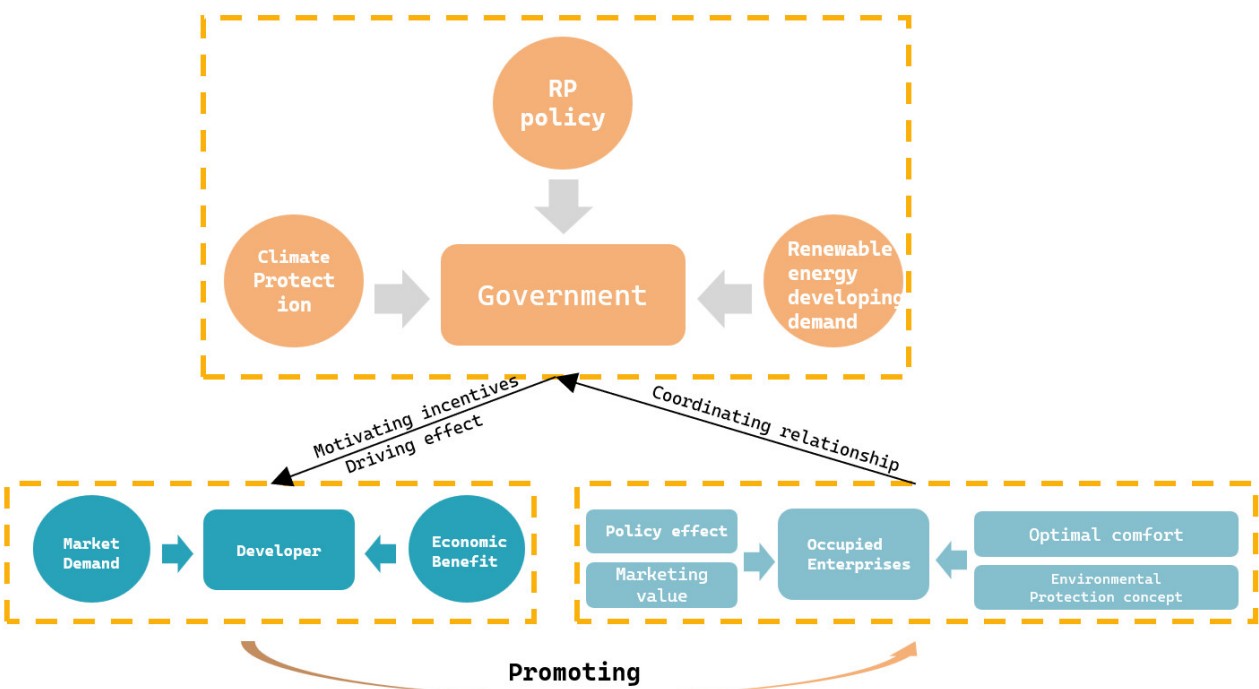

**Figure 5.** Interactive relationship between targeted agents. (Note: RP refers to regulation-based policy.)

**Assumption 1.** *In this model, the beneficial agents are limited to the range of governments, developers, and occupants, and all agents are limited and rational with an ability to assess limited information. In addition, the decision-making processes in this model are affected by personal preferences and the level of gained information. Governments can calculate limited financial information of developers, such as cost, benefit, and cash flow.*

**Assumption 2.** *In this model, the government can employ two strategies (to regulate or not to regulate) for the green retrofitting process, which can be signed as an assemblage (to regulate or not to regulate) and noted as $G = (G_1, G_2)$. Strategies for commercial building developers (real-estate property management enterprises) can be signed to promote green retrofitting or to not promote green retrofitting, and are signed as an assemblage (to promote or not to promote) and noted as $D = (D_1, D_2)$. Occupied merchants can choose to accept the green retrofitting or not to accept, and can be signed as an assemblage (to accept or not to accept) and noted as $O = (O_1, O_2)$.*

**Assumption 3.** *In this model, occupants will consider the comprehensive economic benefit and loss due to green retrofit on both single quantitative and social sides. The occupants will receive additional energy savings due to green retrofitting and social benefits, as these promote thermal comfort and green marketing value. In addition, occupants may incur economic loss due to innovation retrofitting during the construction process.*

**Assumption 4.** *In this model, the government can gather more confidence and support from the public and meet various global agreements when developers and occupants support green retrofitting during the regulation process by federal and local governments.*

### 3.2. Model Establishment

Evolutionary game simulates the entire game process by employing three research agents, and by quantifying benefits using a numerical CBA method to suggest the actual benefit gained by agents. In addition, the actual quantitative costs and benefits act as model variables. This research creatively considered social benefits and contract dispute problems by adding those detailed definitions to the variables.

For governments:

$a_1$ is the benefit (mainly tax) when the government chooses not to regulate and push green retrofitting; $a_2$ is the extra benefit when the government stimulates green retrofitting for enlarged social credibility and increased jobs; $p_1$ is the financial-based cost when the government promotes green retrofitting, such as tax and subsidies; $p_2$ is extra management cost for the government's policymaking, promotion of assessment systems, and awareness of green retrofitted building to the public.

For developers (property management companies):

$b_1$ is the benefit when developers choose not to support green retrofitting; $b_2$ is the benefit due to energy saving when developers support green retrofitting by utilizing new technologies, such as polar energy and energy storage systems; $b_3$ is the additional potential benefit when developers support green retrofitting owing to the possible increase in rent and transaction price; $b_4$ is the cost for compensating irregular business of occupied enterprises during retrofitting; $c$ is the punishment fine when green retrofitting is not promoted and existing buildings incur environmental problems when the standard of existing building is enhanced; and $d$ is the additional cost required to develop innovations of existing building mainly for facilities promotion.

For occupants (occupied enterprises):

$c_1$ is the cost of occupant enterprises to rent green retrofitted buildings; $c_2$ is the cost incurred by occupant enterprises when they do not support green retrofitting and decide to rescind the contract with the property manager (developer); $c_3$ is the numerative additional multiple social benefits for occupant enterprises to support and rent green retrofitted building owing to enhanced thermal comfort; $-c_3$ is the direct economic negative loss due to objection to the retrofitting of occupied building owing to unavoidable time cost; $-c_4$ is the indirect economic negative loss due to the physiological disappointment of not supporting green retrofit; $m$ is the penalty ratio from governmental regulation between developers and occupied enterprises.

Strategies set among the government, developers, and occupied enterprises can be noted as eight combinations: $(G_1, O_1, D_1)$; $(G_1, O_2, D_1)$; $(G_1, O_1, D_2)$; $(G_1, O_2, D_2)$; $(G_2, O_1, D_1)$; $(G_2, O_2, D_1)$; $(G_2, O_1, D_2)$; and $(G_2, O_2, D_2)$. The variables and benefits are listed in Tables 3 and 4.

**Table 3.** Definitions of set parameters.

| Parameters | Group | Definition |
|:---:|:---:|:---|
| $a_1$ | Government | financial revenue without regulation of green retrofit |
| $a_2$ | Government | additional benefit when green retrofitting is promoted for enlarged social credibility, improved environment, and increased jobs |
| $p_1$ | Government | subsidy and technological innovation cost when the government promotes green retrofitting |
| $p_2$ | Government | incremental management cost for the government's policymaking |
| $b_1$ | Developer | original revenue from normal operation |
| $b_2$ | Developer | benefit due to energy saving |
| $b_3$ | Developer | additional potential and social benefit due to increased rent and transaction price |
| $b_4$ | Developer | cost for compensating the irregular business of occupied enterprises |
| $b_5$ | Developer | punishment fine when existing buildings incur environmental problems |
| $b_6$ | Developer | additional cost for developing innovations of existing buildings, mainly for promoting facilities |
| $f$ | Occupant | original revenue from normal operation |
| $c$ | Occupant | original cost for occupant enterprises to rent traditional retrofitted building |
| $c_1$ | Occupant | cost when occupant enterprises rent green retrofitted building |

**Table 3.** *Cont.*

| Parameters | Group | Definition |
|---|---|---|
| $c_2$ | Occupant | cost when occupant enterprises do not support green retrofitting and choose to rescind the contract with the property manager (developer) |
| $c_3$ | Occupant | economic negative loss due to the retrofitting of an occupied building due to an unavoidable time cost |
| $c_4$ | Occupant | additional potential and social benefit due to increased thermal comfort and physiological benefit |
| $m$ | Occupant | the penalty ratio from the governmental regulation between developers and occupied enterprises |

**Table 4.** Definitions of set variables.

| Game Strategy | Government Benefit | Developer Benefit | Occupant Benefit |
|---|---|---|---|
| $(G_1, D_1, O_1)$ | $a_1 + a_2 - p_1 - p_2$ | $b_1 + b_2 + p_1 + b_3 - b_6$ | $f + c - c_1 + c_4 + b_4 - c_3$ |
| $(G_1, D_1, O_2)$ | $a_1 + a_2 - p_1 - p_2 + b_5$ | $b_1 + b_2 + p_1 + b_3 - b_6 - (1 - m) b_5 + c_2$ | $f + c_1 - c - c_2 - c_3 - mb_5$ |
| $(G_2, D_1, O_1)$ | $a_1 + a_2$ | $b_1 + b_2 + b_3 - b_6$ | $f + c - c_1 + c_4 - c_3$ |
| $(G_2, D_1, O_2)$ | $a_1 + a_2$ | $b_1 + b_2 + b_3 - b_4 - b_6$ | $f + c_1 - c + b_4 - c_3$ |
| $(G_1, D_2, O_1)$ | $a_1 + a_2 - p_1 - p_2 + b_5$ | $b_1 - b_4 - (1 - m) b_5$ | $f + c - c_1 - c_3 - mb_5 + b_4 + c_4$ |
| $(G_1, D_2, O_2)$ | $a_1 + a_2 - p_2 + b_5$ | $b_1 - (1 - m) b_5$ | $f + c_1 - c - mb_5$ |
| $(G_2, D_2, O_1)$ | $a_1 + a_2$ | $b_1 + c_2$ | $f + c - c_1 - c_2 - c_3 + c_4$ |
| $(G_2, D_2, O_2)$ | $a_1$ | $b_1$ | $f$ |

Governmental policies are basic principles and guidelines for all developers and occupied enterprises when rescinding activity occurs. Therefore, when agents hold different standpoints, the agent holding a different view from the government is set as default to be paid for liquidated damage for the counterpart. After retrofitting the situation and plan of the occupied site, when occupants choose to not support green retrofitting, enterprises must rescind the contract with developers. During the initial evolutionary game stage, we assumed that the probability that the government chooses regular green retrofitting is $x$, and not choosing regular green retrofitting is $1 - x$; for developers, the probability of developing green retrofitting was assumed to be $y$, and not developing was assumed to be $1 - y$; for occupants, the probability of choosing retrofitted building was assumed to be $z$, and choosing traditional building was assumed to be $1 - z$. In summary, in the entire game process, the government's choice of strategies aimed to "encourage green retrofitting and not to encourage green retrofitting", the developer's strategy included "promotion and no promotion", and the occupant's strategy included a "retrofitted building or traditional building". In this study, these strategies were denoted as $x$, $1 - x$, $y$, $1 - y$, $z$, and $1 - z$, respectively, to indicate the probability that the government supervises developers, and the occupants participate in the project.

In this model,

$$x, y, z \in (0, 1) \tag{1}$$

For governments, the expected benefit of "to regulate", "not to regulate", and average return are denoted as $W_1$, $W_2$, and $W_3$, respectively.

$$W_1 = yz (a_1 + a_2 - p_1 - p_2) + y(1 - z) (a_1 + a_2 - p_1 - p_2 + b_5) + z(1 - y) (a_1 + a_2 - p_1 - p_2 + b_5) + (1 - y) (1 - z) (a_1 + a_2 - p_2 + b_5) \tag{2}$$

$$W_2 = yz (a_1 + a_2) + y(1 - z) (a_1 + a_2) + z(1 - y) (a_1 + a_2) + (1 - y) (1 - z) a_1 \tag{3}$$

$$W_3 = xW_1 + (1 - x) W_2 \tag{4}$$

For developers, the expected benefits of promotion, non-promotion, and average return are denoted as $W_4$, $W_5$, and $W_6$, respectively.

$$W_4 = xz\,(b_1 + b_2 + p_1 + b_3 - b_6) + x(1 - z)\,[b_1 + b_2 + p_1 + b_3 - b_6 - (1 - m)\,b_5 + c_2] + z(1 - x)\,(b_1 + b_2 + b_3 - b_6) + (1 - x)\,(1 - z)\,(b_1 + b_2 + b_3 - b_4 - b_6) \tag{5}$$

$$W_5 = xz\,[b_1 - b_4 - (1 - m)\,b_5] + x(1 - z)\,[b_1 - (1 - m)\,b_5] + z(1 - x)\,(b_1 + c_2) + (1 - y)\,(1 - z)\,b_1 \tag{6}$$

$$W_6 = yW_4 + (1 - y)\,W_5 \tag{7}$$

For occupants, the expected benefits of choosing a retrofitted building, traditional building, and the average return are denoted as $W_7$, $W_8$, and $W_9$, respectively.

$$W_7 = xy\,(f + c - c_1 + c_4 + b_4 - c_3) + y(1 - x)\,(f + c - c_1 + c_4 - c_3) + x(1 - y)\,(f + c - c_1 - c_3 - mb_5 + b_4 + c_4) + (1 - x)\,(1 - y)\,(f + c - c_1 - c_2 - c_3 + c_4) \tag{8}$$

$$W_8 = xy\,(f + c_1 - c - c_2 - c_3 - mb_5) + y(1 - x)\,(f + c_1 - c + b_4 - c_3) + x(1 - y)\,(f + c_1 - c - mb_5) + (1 - x)\,(1 - y)\,f \tag{9}$$

$$W_9 = zW_7 + (1 - z)\,W_8 \tag{10}$$

The obtained payoff matrix is illustrated in Table 5.

**Table 5.** Payoff matrix of the evolutionary dynamic model under the government, developers, and occupant behaviors.

| Game Agents | | | | Government | |
|---|---|---|---|---|---|
| | | | | Regulation (X) | Non-Regulation (1 − X) |
| Developer | promotion (y) | Occupant | retrofit (z) | $a_1 + a_2 - p_1 - p_2, b_1 + b_2 + p_1 + b_3 - b_6, f + c - c_1 + c_4 + b_4 - c_3$ | $a_1 + a_2, b_1 + b_2 + b_3 - b_6, f + c - c_1 + c_4 - c_3$ |
| | | | tradition (1 − z) | $a_1 + a_2 - p_1 - p_2 + b_5, b_1 + b_2 + p_1 + b_3 - b_6 - (1 - m)\,b_5 + c_2, f + c_1 - c - c_2 - c_3 - mb_5$ | $a_1 + a_2, b_1 + b_2 + p_1 + b_3 - b_6, b_1 + b_2 + b_3 - b_4 - b_6, f + c_1 - c + b_4 - c_3$ |
| | non-promotion (1 − y) | | retrofit | $a_1 + a_2 - p_1 - p_2 + b_5, b_1 - b_4 - (1 - m)\,b_5, f + c - c_1 - c_3 - mb_5 + b_4 + c_4$ | $a_1 + a_2, b_1 + c_2, f + c - c_1 - c_2 - c_3 + c_4$ |
| | | | tradition (1 − z) | $a_1 + a_2 - p_2 + b_5, b_1 - (1 - m)\,b_5, f + c_1 - c - mb_5$ | $a_1, b_1, f$ |

### 3.3. Model Analysis

### 3.3.1. CBA of Government

The replicator dynamic equation obtained by analyzing the benefit of the government during the game is expressed as follows:

$$W_1 = a_1 + a_2 + b_5 - p_2 - p_1 y - p_1 z - b_5 yz + p_1 yz \tag{11}$$

$$W_2 = xz\,[b_1 - b_4 - (1 - m)\,b_5] + x(1 - z)\,[b_1 - (1 - m)\,b_5] + z(1 - x)\,(b_1 + c_2) + (1 - y)\,(1 - z)\,b_1 \tag{12}$$

$$W_3 = xW_1 + (1 - x)\,W_2 = a_1 + a_2 x + a_2 y + b_5 x + a_2 z - p_2 x - a_2 xy - a_2 xz - a_2 yz - p_1 xy - p_1 xz + a_2 xyz - b_5 xyz + p_1 xyz \tag{13}$$

$$F_1(x) = dx/dt = x\,(W_1 - W_3) = x(1 - x)\,(a_2 + b_5 - p_2 - a_2 y - a_2 z - p_1 y - p_1 z + a_2 yz - b_5 yz + p_1 yz)$$

$$= x(1 - x)\,[(a_2 z - p_1 - a_2 - b_5 z + p_1 z)\,y + a_2 + b_5 - p_2 - a_2 z - p_1 z)]$$

$$X \times (1 - x) \times ((a_2 \times z - p_1 - a_2 - b_5 \times z + p_1 \times z) \times y + a_2 + b_5 - p_2 - a_2 \times z - p_1 \times z)]$$



By estimating the derivation of $F(x)$, we obtain:

$$F_1(x)' = dF_1(x)/dx = (2x - 1)(a_2 + b_5 - p_2 - a_2 y - a_2 z - p_1 y - p_1 z + a_2 yz - b_5 yz + p_1 yz) \quad (14)$$

When $F_1(x) = 0$ and $F_1(x)' < 0$, the government group attains its evolutionary stability strategy (ESS) point. As the evolutionary stability strategy (ESS) point of the government group can be expressed as $x_1 = 0$, $x_2 = 1$, or $you = y_0 = [z(a_2 + p_1) + p_2 - a_2 - a_5]/[z(a_2 + b_5 + p_1) - p_1 - p_2]$, there are three possible scenarios:

(1) When $you = y_0 = [z(a_2 + p_1) + p_2 - a_2 - a_5]/[z(a_2 + b_5 + p_1) - p_1 - p_2]$, if $F_1(x) = 0$, the evolutionary strategy is stable regardless of the change in the value of $x$, indicating that when the probability that the developer is willing to promote green retrofitting is equal to $y_0 = z(a_2 + p_1) + p_2 - a_2 - a_5]/[z(a_2 + b_5 + p_1) - p_1 - p_2]$, the benefits offered by the government for both regulation and non-regulation strategies are equal.

(2) When $y > y_0 = y_0 = [z(a_2 + p_1) + p_2 - a_2 - a_5]/[z(a_2 + b_5 + p_1) - p_1 - p_2]$, if $F_1(x) = 0$, $x_1 = 0$ and $x_2 = 1$ are the two ESS points of the game process. $F_1(0)' < 0$, $F_1(1)' > 0$, $x_1 = 0$ is the evolutionary stable strategy of the government, indicating that when the probability that the developers will support green retrofitting promotion is greater than $[z(a_2 + p_1) + p_2 - a_2 - a_5/z(a_2 + b_5 + p_1) - p_1 - p_2]$, the strategy of the government changes from regulation to non-regulation, and non-regulation becomes the stable strategy.

(3) When $y < y_0 = y_0 = z(a_2 + p_1) + p_2 - a_2 - a_5]/[z(a_2 + b_5 + p_1) - p_1 - p_2]$, if $F_1(x) = 0$, $x_1 = 0$ and $x_2 = 1$ are the two ESS points of the game process. $F_1(0)' < 0$, $F_1(1)' > 0$, $x_2 = 1$ is the evolutionary stable strategy of the government, indicating that when the probability of developers supporting green retrofitting promotion is lower than $[z(a_2 + p_1) + p_2 - a_2 - a_5/z(a_2 + b_5 + p_1) - p_1 - p_2]$, the strategy of the government changes from non-regulation to regulation, and regulation becomes the stable strategy.

### 3.3.2. CBA of Developers

The replicator dynamic equation of the government obtained while analyzing the benefit of developers during the game is shown as follows:

$$\begin{aligned} W_4 &= xz(b_1 + b_2 + p_1 + b_3 - b_6) + x(1 - z)[b_1 + b_2 + p_1 + b_3 - b_6 - (1 - m)b_5 + c_2] + \\ & z(1 - x)(b_1 + b_2 + b_3 - b_6) + (1 - x)(1 - z)(b_1 + b_2 + b_3 - b_4 - b_6) = b_1 + b_2 + b_3 - b_4 - b_6 \\ & + b_4 x - b_5 x + c_2 x + b_4 z + p_1 x + b_5 mx - b_4 xz + b_5 xz - c_2 xz - b_5 mxz \end{aligned} \quad (15)$$

$$\begin{aligned} W_5 &= xz[b_1 - b_4 - (1 - m)b_5] + x(1 - z)[b_1 - (1 - m)b_5] + z(1 - x)(b_1 + c_2) + (1 - x) \\ & (1 - z)b_1 = b_1 - b_5 x + c_2 z + b_5 mx - b_4 xz - c_2 xz \end{aligned} \quad (16)$$

$$\begin{aligned} W_6 &= yW_4 + (1 - y)W_5 = b_1 - b_5 x + b_2 y + b_3 y - b_4 y - b_6 y + c_2 z + b_5 mx + b_4 xy + c_2 xy \\ & - b_4 xz - c_2 xz + b_4 yz - c_2 yz + p_1 xy + b_5 xyz - b_5 mxyz \end{aligned} \quad (17)$$

$$\begin{aligned} F_2(y) &= dy/dt = y(W_4 - W_6) = y(1 - y)(b_2 + b_3 - b_4 - b_6 + b_4 x + c_2 x + b_4 z - c_2 z + p_1 x + \\ & b_5 xz - b_5 mxz) = y(1 - y)[(b_4 + c_2 + p_1 + b_5 z - b_5 mz)x + b_2 + b_3 - b_4 - b_6 + b_4 z - c_2 z)] \end{aligned}$$

The estimation of the derivation of $F_2(y)$ can be expressed as :

$$F_2(y)' = dF_2(x)/dy = (1 - 2y)(b_2 + b_3 - b_4 - b_6 + b_4 x + c_2 x + b_4 z - c_2 z + p_1 x + b_5 xz - b_5 mxz) \quad (18)$$

When $F_2(y) = 0$ and $F_2(y)' < 0$, the developer group attains its ESS point. As the ESS points of the government group are $y_1 = 0$, $y_2 = 1$, or $x_0 = [b_2 + b_3 - b_4 - b_6 + b_4 z - c_2 z]/[b_4 + c_2 + p_1 + b_5 z - b_5 mz]$, there are three possible scenarios:

(1) When $x = x_0 = [b_2 + b_3 - b_4 - b_6 + b_4 z - c_2 z]/[b_4 + c_2 + p_1 + b_5 z - b_5 mz]$

When $F_2(y) = 0$, the evolutionary strategy is stable regardless of the change in the value of $y$, indicating that when the probability that the government is willing to promote green retrofitting is equal to $x_0$, the benefits offered by the government in both the promotion and non-promotion strategies are equal.

(2) When $x > x_0 = [b_2 + b_3 - b_4 - b_6 + b_4z - c_2z]/[b_4 + c_2 + p_1 + b_5z - b_5mz]$, if $F_2(y) = 0$, $y_1 = 0$ and $y_2 = 1$ are the two ESS points of the entire game process. $F_2(0)' > 0$, $F_2(1)' < 0$, $y_1 = 1$ is the evolutionary stable strategy of the developer, indicating that when the probability of the government regulating green retrofitting is greater than $[b_2 + b_3 - b_4 - b_6 + b_4z - c_2z/b_4 + c_2 + p_1 + b_5z - b_5mz]$, the strategy of the developer changes from non-promotion to promotion, and promotion becomes the stable strategy.

(3) When $x < x_0 = [b_2 + b_3 - b_4 - b_6 + b_4z - c_2z]/[b_4 + c_2 + p_1 + b_5z - b_5mz]$, if $F_2(y) = 0$, $y_1 = 0$ and $y_2 = 1$ are the two ESS points of the game process. $F_2(0)' < 0$, $F_2(1)' > 0$, $x_2 = 0$ is the evolutionary stable strategy of the government, indicating that when the probability that the government regulating green retrofitting promotion is greater than $[b_2 + b_3 - b_4 - b_6 + b_4z - c_2z/b_4 + c_2 + p_1 + b_5z - b_5mz]$, the strategy of the developer changes from promotion to non-promotion, and non-promotion becomes the stable strategy.

### 3.3.3. CBA of Occupied Enterprises

The replicator dynamic equation of the occupied enterprises obtained during the analysis of the benefit of occupied enterprises during the game is expressed as follows:

$$W_7 = xy\,(f + c - c_1 + c_4 + b_4 - c_3) + y(1 - x)\,(f + c - c_1 + c_4 - c_3) + x(1 - y)\,(f + c - c_1 - c_3 - mb_5 + b_4 + c_4) + (1 - x)\,(1 - y)\,(f + c - c_1 - c_2 - c_3 + c_4) \tag{19}$$

$$W_8 = xy\,(f + c_1 - c - c_2 - c_3 - mb_5) + y(1 - x)\,(f + c_1 - c + b_4 - c_3) + x(1 - y)\,(f + c_1 - c - mb_5) + (1 - x)\,(1 - y)\,f = f - cx + c_1x + b_4y - cy + c_1y - c_3y - b_5mx - b_4xy + cxy - c_1xy - c_2xy \tag{20}$$

$$W_9 = zW_7 + (1 - z)\,W_8 = f - cx + c_1x + b_4y - cy + c_1y - c_3y + cz - c_1z - c_2z - c_3z + c_4z - b_5mx - b_4xy + cxy - c_1xy - c_2xy + b_4xz + cxz - c_1xz + c_2xz - b_4yz + cyz - c_1yz + c_2yz + c_3yz + b_4xyz - cxyz + c_1xyz + b_5mxyz \tag{21}$$

$$F_3(z) = dz/dt = z\,(W_7 - W_9) = z(1 - z)\,(c - c_1 - c_2 - c_3 + c_4 + b_4x + cx - c_1x + c_2x - b_4y + cy - c_1y + c_2y + c_3y + b_4xy - cxy + c_1xy + b_5mxy) = z(1 - z)\,[(c - b_4 - c_1 + c_2 + c_3 + b_4x - cx + c_1x + b_5mx)\,y + c - c_1 - c_2 - c_3 + c_4 + b_4x + cx - c_1x + c_2x]$$

The estimation of the derivation of $F(x)$ can be expressed as:

$$F_3(z)' = dF_3(z)/dz = (2z - 1)\,(c - c_1 - c_2 - c_3 + c_4 + b_4x + cx - c_1x + c_2x - b_4y + cy - c_1y + c_2y + c_3y + b_4xy - cxy + c_1xy + b_5mxy) \tag{22}$$

When $F_3(z) = 0$ and $F_3(z)' < 0$, the government group attains its ESS point. As the ESS points of the government group are $x_1 = 0$, $x_2 = 1$, *or* $y_0 = [c - c_1 - c_2 - c_3 + c_4 + b_4x + cx - c_1x + c_2x]/[c - b_4 - c_1 + c_2 + c_3 + b_4x - cx + c_1x + b_5mx]$, there are three possible scenarios:

(1) When $y = y_0 = [c - c_1 - c_2 - c_3 + c_4 + b_4x + cx - c_1x + c_2x]/[c - b_4 - c_1 + c_2 + c_3 + b_4x - cx + c_1x + b_5mx]$, if $F_3(x) = 0$, the evolutionary strategy is stable regardless of the change in the value of $x$, indicating that when the probability that the developer is willing to promote green retrofitting is equal to $[c - c_1 - c_2 - c_3 + c_4 + b_4x + cx - c_1x + c_2x/c - b_4 - c_1 + c_2 + c_3 + b_4x - cx + c_1x + b_5mx]$, the benefit offered by the government in both regulation and non-regulation strategies is equal.

(2) When $y > y = [c - c_1 - c_2 - c_3 + c_4 + b_4x + cx - c_1x + c_2x]/[c - b_4 - c_1 + c_2 + c_3 + b_4x - cx + c_1x + b_5mx]$, if $F_3(z) = 0$, $y_1 = 0$ and $y_2 = 1$ are the two ESS points of the game process. $F_3(0)' < 0$, $F_3(1)' > 0$, $x_1 = 0$ is the evolutionary stable strategy of the government, indicating that when the probability that the developers will support the green retrofitting promotion is greater than $[c - c_1 - c_2 - c_3 + c_4 + b_4x + cx - c_1x + c_2x/c - b_4 - c_1 + c_2 + c_3 + b_4x - cx + c_1x + b_5mx]$, the strategy of the occupied enterprises changes from regulation to non-regulation, and non-regulation becomes the stable strategy.

(3) When $y < y_0 = [c - c_1 - c_2 - c_3 + c_4 + b_4x + cx - c_1x + c_2x]/[c - b_4 - c_1 + c_2 + c_3 + b_4x - cx + c_1x + b_5mx]$, if $F_3(z) = 0$, $y_1 = 0$ and $y_2 = 1$ are the two ESS points of the game process. $F_3(0)' < 0$, $F_3(1)' > 0$, $y_2 = 1$ is the evolutionary stable strategy of the government, indicating that when the probability that the developers will support green retrofitting

promotion is greater than $[c - c_1 - c_2 - c_3 + c_4 + b_4x + cx - c_1x + c_2x/c - b_4 - c_1 + c_2 + c_3 + b_4x - cx + c_1x + b_5mx]$, the strategy of occupied enterprises changes from non-regulation to regulation, and regulation becomes the stable strategy.

### 3.4. Strategy Analysis of the Multiple-Agent Evolutionary Dynamic Process

The stability of the equilibrium point can be observed by analyzing the Jacobian matrix, when the situation meets $det(J) > 0$ and $tr(J) < 0$, generally, the stable point is the ESS point [10]. The calculations of the determinant and trace of the Jacobian matrix are expressed below:

$$\text{det.j} = \frac{\partial F(x)}{\partial x} \times \frac{\partial F(y)}{\partial y} - \frac{\partial F(x)}{\partial y} \times \frac{\partial F(y)}{\partial x} \tag{23}$$

$$\text{tr} = \frac{\partial F(x)}{\partial x} + \frac{\partial F(y)}{\partial y} \tag{24}$$

$$J = \begin{bmatrix} \partial F(x)/\partial x & \partial F(x)/\partial x & \partial F(x)/\partial x \\ \partial F(y)/\partial x & \partial F(y)/\partial y & \partial F(y)/\partial y \\ \partial F(z)/\partial x & \partial F(z)/\partial z & \partial F(z)/\partial z \end{bmatrix} \tag{25}$$

$$J = \begin{bmatrix} J_{11} & J_{12} & J_{13} \\ J_{21} & J_{22} & J_{23} \\ J_{31} & J_{32} & J_{33} \end{bmatrix} \tag{26}$$

Among them

$$J_{11} = (2x - 1)(p_2 - b_5 - a_2 + a_2y + a_2z + p_1y + p_1z - a_2yz + b_5yz - p_1yz) \tag{27}$$

$$J_{21} = y(1 - y)(b_4 - b_1 + c_2 + p_1 + b_1z + b_5z - b_5mz) \tag{28}$$

$$J_{21} = y(1 - y)(b_4 - b_1 + c_2 + p_1 + b_1z + b_5z - b_5mz) \tag{29}$$

$$J_{31} = z(1 - z)(b_4 + c - c_1 + c_2 + b_4y - cy + c_1y + b_5my) \tag{30}$$

$$J_{12} = x(x - 1)(a_2 + p_1 - a_2z + b_5z - p_1z) \tag{31}$$

$$J_{22} = b_2 + b_3 - b_4 \quad \begin{aligned} &-b_6 - b_1x + b_4x + 2b_1y - 2b_2y + c_2x - 2b_3y + 2b_4y + 2b_6y + b_4z - c_2z \\ &+p_1x - 3b_1y^2 + 3b_1y^2z + 2b_1xy - 2b_4xy + b_1xz - 2c_2xy + b_5xz - 2b_1yz \\ &-2b_4yz + 2c_2yz - 2p_1xy - b_5mxz - 2b_1xyz - 2b_5xyz + 2b_5mxyz \end{aligned} \tag{32}$$

$$J_{32} = z(1 - z)(c - b_4 - c_1 + c_2 + c_3 + b_4x - cx + c_1x + b_5mx) \tag{33}$$

$$J_{13} = x(x - 1)(a_2 + p_1 - a_2y + b_5y - p_1y) \tag{34}$$

$$J_{23} = y(1 - y)(b_4 - c_2 + b_1x + b_5x - b_1y - b_5mx) \tag{35}$$

$$J_{33} = (1 - 2z)(c - \quad c_1 - c_2 - c_3 + c_4 + b_4x + cx - c_1x + c_2x - b_4y + cy - c_1y + c_2y + c_3y \\ +b_4xy - cxy + c_1xy + b_5mxy) \tag{36}$$

By bringing data into the Jacobian matrix after calculation, the eight equilibrium points, which are the pure strategy solutions, can be obtained sequentially. Table 6 illustrates the obtained eigenvalues of each equilibrium point. Points whose $\lambda_1$, $\lambda_2$, $\lambda_3$ character of equilibrium points are all less than zero are considered as the ESS points where plus or minus characteristics can be obtained by manual determination.

**Table 6.** Condition of the evolutionary stability of the equilibrium points.

| Equilibrium Point | | Feature Values | | |
|---|---|---|---|---|
| | | $\lambda_1$ | $\lambda_2$ | $\lambda_3$ |
| $E_1$ | $(0,0,0)$ | $a_2 + b_5 - p_2$ | $b_2 + b_3 - b_4 - b_6$ | $c - c_1 - c_2 - c_3 + c_4$ |
| $E_2$ | $(0,0,1)$ | $b_5 - p_1 - p_2$ | $b_2 + b_3 - b_6 - c_2$ | $c_1 - c + c_2 + c_3 - c_4$ |
| $E_3$ | $(0,1,0)$ | $b_5 - p_1 - p_2$ | $2c - b_4 - 2c_1 + c_4$ | $b_4 - b_2 - b_3 - b_1 + b_6$ |
| $E_4$ | $(0,1,1)$ | $-p_1 - p_2$ | $b_4 - 2c + 2c_1 - c_4$ | $b_6 - b_3 - b_2 + c_2$ |
| $E_5$ | $(1,0,0)$ | $p_2 - b_5 - a_2$ | $b_4 + 2c - 2c_1 - c_3 + c_4$ | $b_2 - b_1 + b_3 - b_6 + c_2 + p_1$ |
| $E_6$ | $(1,0,1)$ | $p_1 - b_5 + p_2$ | $2c_1 - 2c - b_4 + c_3 - c_4$ | $b_2 + b_3 + b_4 + b_5 - b_6 + p_1 - mb_5$ |
| $E_7$ | $(1,1,0)$ | $p_1 - b_5 + p_2$ | $b_6 - b_3 - b_2 - c_2 - p_1$ | $b_4 + 2c - 2c_1 + c_2 + c_4 + b_5 m$ |
| $E_8$ | $(1,1,1)$ | $p_1 + p_2$ | $2c_1 - 2c - b_4 - c_2 - c_4 - b_5 m$ | $b_6 - b_3 - b_4 - b_5 - b_2 - p_1 + mb_5$ |

As shown in Table 6, parameters were set at values higher than zero; thus, $p_1 + p_2$ are higher than zero and $E_8(1,1,1)$ is unstable. Specific judgement is essential as the stabilities of the other seven points are not certain. Thus, to obtain $E_5(1,0,0)$, in the initial stage of green retrofitting, owing to the decisive and macroscopical regulation character of the Chinese government and the pressure of global green agreements, the government tends to strongly push the green retrofitting process. However, this results in a significant increase in the management and subsidy cost. Powerful regulation with immature, unqualified, and non-awareness-promoting market results in high pressure and difficulties for the developer and occupied enterprises. Therefore, developers can choose not to develop green promotion, and enterprises can choose not to support green retrofitting and support traditional building occupancy. In contrast, with the development and maturity of the human resource market qualifying process, developers and enterprises are aware of green retrofitting and have perceived the increasing comfort and benefit of retrofitted buildings. This ensures that the market-oriented mechanism forms the main body of the entire multiple relationships, which correspond to $E_4 (0,1,1)$.

In a situation where the $\lambda$ characters of the points are all less than zero, the point can be considered as an ESS point. As an example, $E_5(1,0,0)$ is further explained below.

In the first inequality:

If $p_2 < b_5 + a_2$, the management cost of the government is less than the income from the punishment of developers and occupied enterprises and social benefit from the government.

If $b_4 + 2c + c_4 < 2c_1 + c_3$, the value that enterprises obtain from increasing thermal comfort and the business loss compensation provided by developers with traditional occupancy loss is still less than the promoted situation with business loss.

If $b_2 + b_3 + c_2 + p_1 < b_1 + b_6$, the incomes from energy savings, social benefit, compensation, and subsidies from governments are less than the original benefit and promotion cost.

In this situation, developers and enterprises choose negative behaviors. Similarly, other points can be explained when this point is illustrated as an ESS point.

## 4. Numerical Simulation and Discussion

### 4.1. Multiple-Agent Evolutionary Game Results

The study analyzed the behavior selection mechanism of government, developers, and occupants' merchants under different scenarios caused by the complexity of parameters and inconvenience for specific analysis. In this study, numerical simulation and verification of the evolutionary game model were performed using MATLAB 2021b software programming. The above analysis expressed in the previous section indicated that the parameters significantly affected the dynamic evolutionary process.

To meet the benefit demand and obtain rational and ideal simulation results, the initial parameters were set as follows: $a_1 = 30$, $a_2 = 20$, $p_1 = 8$, $p_2 = 4$, $b_1 = 22$, $b_2 = 8$ $b_3 = 5$, $b_4 = 2$, $b_5 = 3$, $b_6 = 18$, $c = 6$, $c_1 = 8$, $c_2 = 2$, $c_3 = 2$, $c_4 = 2$, $m = 0.7$, $f = 12$. Figure 6 shows the simulated result obtained using the set parameters. When the initially set parameters

were used and the probabilities of positive behavior were less than 0.5, the multiple group behaviors approached $E_4(1,0,0)$, indicating that only the governments are pushing green retrofitting via regulation, whereas the developers and enterprises are unwilling to promote it. When the probabilities of positive behavior were above 0.5, the multiple group behaviors approached $E_3(0,1,0)$, indicating that only the developers are willing to develop green retrofitting in this dynamic process. Furthermore, the probability value varied around $p = 0.5$, indicating that 0.5 is the threshold value of the dynamic process, which is the point where the active party changes from the government to the developer owing to the increased punishment to the supervised parties, as well as the overburden of welfare and subsidy costs on the government. For the government, the welfare subsidy is the major factor affecting the entire green retrofitting process as it is a significantly powerful decisive leadership factor of the Chinese government. Accordingly, the willingness and efficiency of the green retrofitting process increased with an increase in the achievable welfare parameter. Figure 7 shows the simulated evolutionary game result with an increase in $p_1$ (i.e., governmental welfare) to 15. Accordingly, when the subsidy parameter value was doubled, the pressure from the government on the developer increased and the financial overburden from the cancellation cost to enterprises increased.

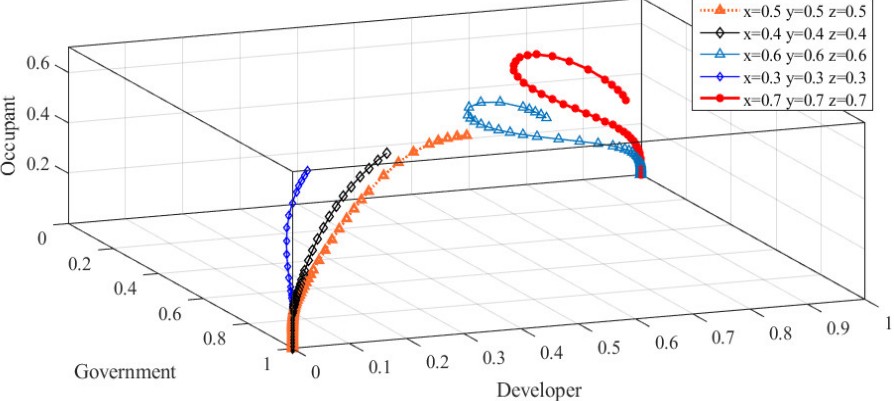

**Figure 6.** Evolutionary game process of the dynamic, increasing the *x*, *y*, and *z* probability using the initial set parameters.

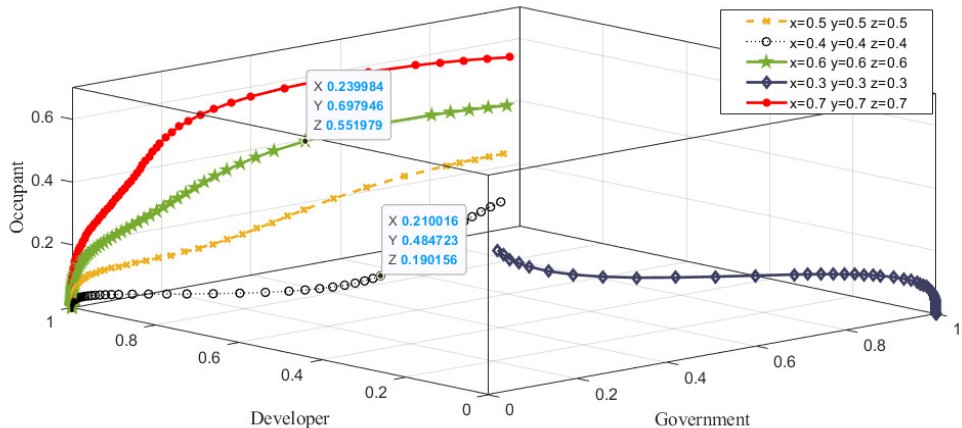

**Figure 7.** Evolutionary game process of the dynamic, the increasing *x*, *y*, and *z* probability when the $p_1$ parameter is doubled.

Compared to when the initial parameters were used, the evolutionary curve of the dynamic strategies approached stability at a higher ratio when the $p_1$ parameter was doubled.

### 4.2. Impact of Different Initial Strategies and Sensitivity Analysis

All the equilibrium points analyzed in this research are limited owing to length limitations. The above analysis of $E_8(1,1,1)$, which is considered the developing stage of the three-class industry developing process, was confirmed to be invalid, which could be attributed to the fact that the Jacobian matrix condition could not be matched. During retrofitting activity, the stable and advanced situation exhibited a linear behavior as it approached the ideal situation, which is $E_5(0,1,1)$.

Figure 8 shows the time evolutionary trajectory of retrofitted buildings with varied governmental financial welfares. Figure 8a, which shows a scenario when the financial welfare is more than 11, indicates that the probabilities of the government motivating green retrofitting exhibits a declining tendency towards non-regulation. Particularly, the government chooses positive behavior when the financial burden due to welfare is approximately less than 50% of the general benefit. Developers directly gain financial welfare from the governmental side and benefit from saving energy, but assume more environmental responsibility than enterprises. Thus, the costs of green retrofitting, such as HVAC and lighting systems, are considerable. Gaining welfare from the government side acts as the main motivation for developers to promote green retrofitting and to investigate the future of regulating government tendencies.

**(a)** Government side

The different incentive intention of the financial welfare of the government, and a high incentive economic incentive can effectively motivate intermediary and terminal agents, but may simultaneously result in a counteracting reaction. When the expected gain by the government is less than the prospective scenario, the government chooses a positive behavior. The behavior of the government exhibits a declining tendency in the initial stage of the process, which may be attributed to the instability in the transitory unbalanced fiscal pressure.

**(b)** Developer side

In contrast, for the developer side, an increase in the financial subsidy results in notable diversity. The most feasible ideal situation for the government and developer to promote green retrofit was observed when $p_1 =$ 11. When $p_1$ is above 9, the government choice generally approaches a non-regulation behavior. However, an increase in the welfare effectively stimulates developers to broaden retrofitting scale.

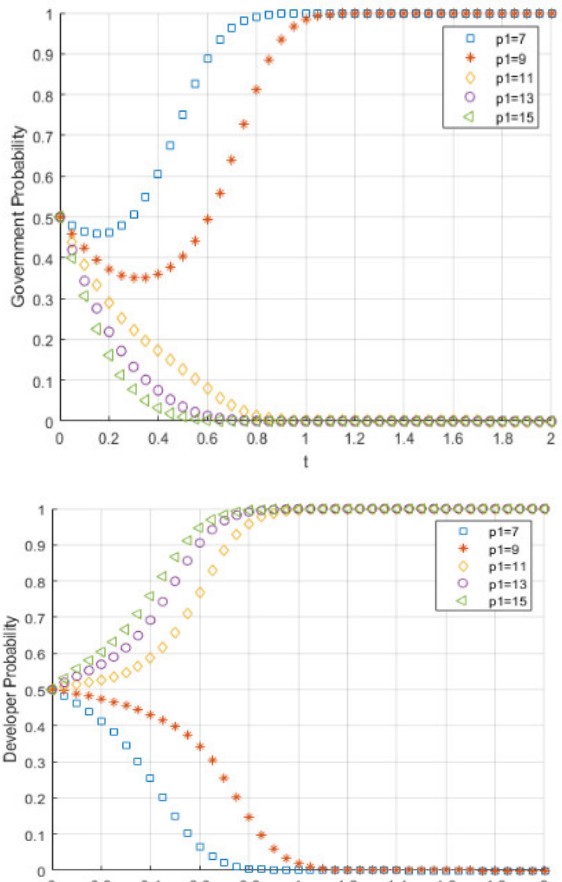

**Figure 8.** *Cont.*

**(c)** Occupant side

Occupants are unlikely to demonstrate a positive choosing tendency under varying parameter situations.

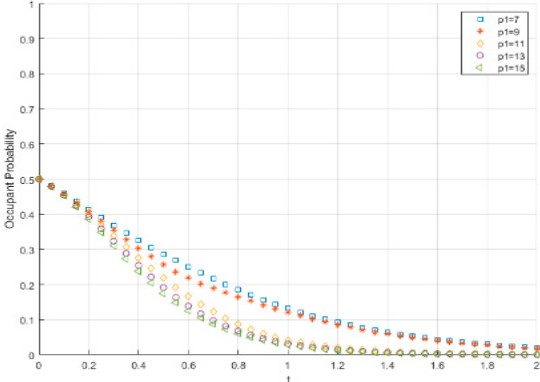

**Figure 8.** Time evolutionary trajectory of retrofitted building with varying governmental financial welfares ($p_1$).

The results in the images indicate that occupied enterprises are vulnerable compared to government and developer agents. A retrofitted business environment provides limited benefits for enterprises, whereas the probability of future economic loss and forfeited value indicates high financial pressure. Developers gain vital punishment and benefits, including economic energy saving and increased social reputation, which may represent a low and non-ideal motivation for occupants (Figures 7–9). Consequently, when owing more slack retrofitting environment for enterprises, in this study, the cost parameters of occupants were re-adjusted to analyze the developing probabilities of occupant agents in this multiple-participants game. When the punishment ratio for developers is reduced, and the incremental rent of real-estate cost and cancellation cost is reduced, the re-combined evolutionary performance of occupants is illustrated below.

As punishment is another powerful tool for regulating agent to manage and promote new policy, governments can motivate a more effective green retrofitting process mainly via welfare and economical punishment ($b_5$). Figure 10 shows various punishment scenarios under an optimized re-adjusted environment situation.

**(a)** Government side

With a change in the punishment value from 1 to 3.5, the government exhibits a similar decision-making behavior tendency, that is, probabilities decrease within a short time and increase until stability. This indicates that punishment income cannot act as the main benefit factor for the government, as varying the punishment values had no significant effect on the governmental behavior.

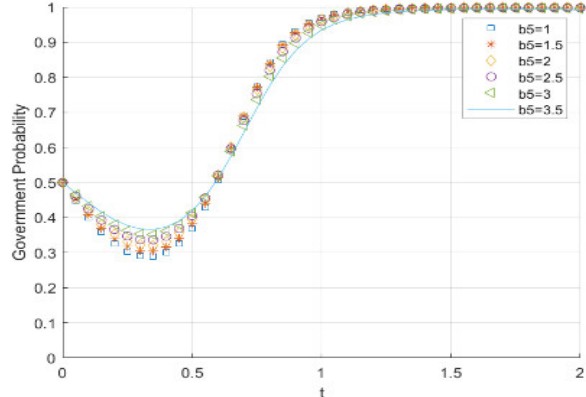

**Figure 9.** *Cont.*

**(b)** Developer side
Compared to that in the initially set environment, developers suffer more strict conditions in the environment optimized for enterprises, exhibiting an opposite trend compared to the government and attaining stability within a similar time towards negative behavior

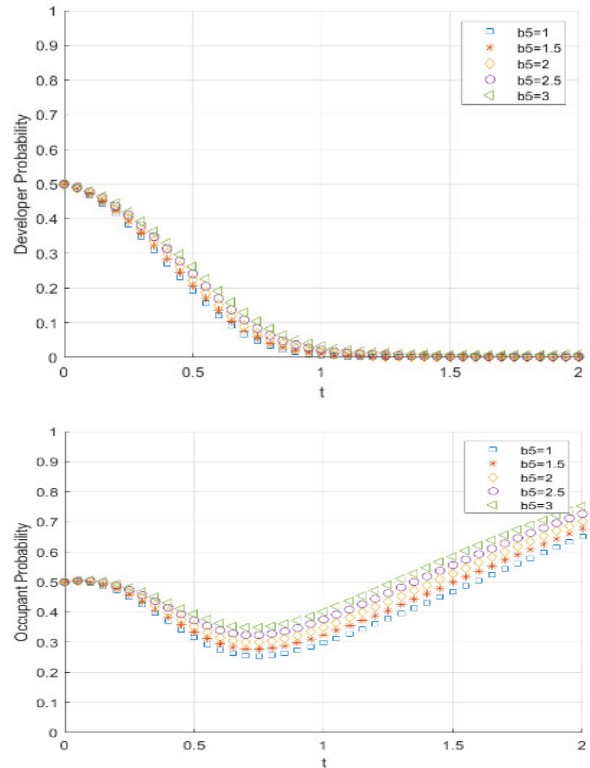

**(c)** Occupant side
Compared to all the aforementioned scenarios, occupants exhibited a reversed (i.e., positive) performance when the environment was optimized for occupants, wherein a low-risk-taking and high-impact punishment is observed among occupants.

**Figure 9.** Time evolutionary trajectory in an optimized occupant environment under varying punishment values ($b_5$ *with adjusted* $c_1$–$c_4$ *condition*).

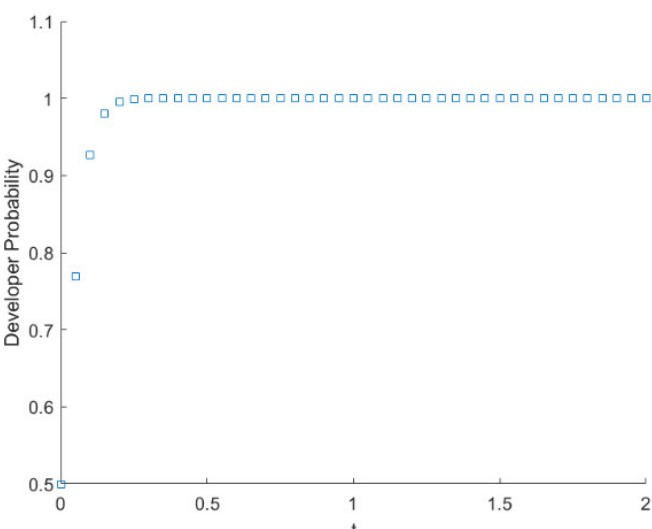

**Figure 10.** Simulated result of one adjusted life-cycle calculated scenario.

*4.3. Life-Cycle Perspective Calibrated Cost-Benefit Evolutionary Sensitivity Analysis*

Typically, comparative high initial cost contributes to a low promotion of the green retrofitting process owing to the high financial pressure on developers. Nevertheless, the ability of optimized buildings to promote energy efficiency performance is frequently ignored. Current evolutionary game-based research has focused on setting investment cost parameters from an initial cost perspective, and has failed to apply the life-cycled calibration method in setting parameters. However, when utilizing the life-cycle perspective method to analyze retrofitting activity, most retrofit investment options have been confirmed to be worthwhile. According to Lu, Li, Lee, and Song [10], higher set-point temperature and lower plug load, occupancy sensor for lighting, efficient water-cooled chiller and green

roof, and their deuteronic combined scenario and measures are cost-effective. Typically, the re-joined annual saving from energy renovated building is higher than the annual cost. However, when adjusting the energy retrofit investment parameters in the LCC method, the values can be calculated using the following formulas:

$$PV_{(Benefit)} = \sum_{t=0}^{N} \frac{CF_t(benefit)}{(1 + i_t)^t} \tag{37}$$

$$PV_{(cost)} = \sum_{t=0}^{N} \frac{CF_t(cost)}{(1 + i_t)^t} \tag{38}$$

$$NPV = PV_{(Benefit)} - PV_{(cost)} \tag{39}$$

$$BCR = \frac{\left|PV_{(Benefit)}\right|}{\left|PV_{(cost)}\right|} \tag{40}$$

where:

$PV$ = the present value;

$CF$ = the cash flow of a period;

$i$ = the discount rate of interest rate;

$N$ = the total number of periods

$t$ = the occurence periods;

$\Delta E$ = annual energy saving.

Retrofitted green buildings can considerably save costs owing to the increase in energy efficiency, whereas previous studies have revealed that individual retrofit construction, such as the individual construction of an HVAC system, building envelope renovation, water retention improvement, and renewable energy utilization, is only capable of saving 11.75, 30.19, 19.57, 15.22, and 4.35%, respectively, of the total energy usage [85–88]. The intermediary position of developers can be mainly attributed to technical challenges. Lu et al. [78] demonstrated detailed realistic data information on integrated green techniques through various energy retrofit options. Using the data obtained by Lu et al. [78] in their case study, and setting the occurring period as twenty years, we summarized the annual savings and benefits and the calculated ratio of saving and benefit of the aforementioned measures. Hence, based on the initial set parameter matrix ($b_2$ = 8, $b_6$ = 18), the adjusted energy saving ($b_2$) and investment cost parameter ($b_6$) are collectively summarized in Table 7:

**Table 7.** Referred cost benefit of the retrofitted case and adjusted parameters.

| Options | Annual Cost/$ | Annual Saving/$ | Ratio | Adjusted Parameters | |
|---|---|---|---|---|---|
| | | | | $b_2$ | $b_6$ |
| Higher set-point temperature and lower plug load | 9250 | 23,818 | 2.58 | 21 | 8 |
| Occupancy sensor for lighting | 1932 | 12,737 | 7.10 | 28 | 4 |
| Combined scenario | 11,182 | 61,213 | 5.48 | 137 | 25 |
| Efficient water-cooling chiller | 28,297 | 41,839 | 1.49 | 22 | 14 |
| Green roof | 4701 | 22,433 | 4.77 | 20 | 7 |

As shown in Figure 10, in the adjusted investment parameter condition, five scenarios with a tendency for the developers to choose a positive behavior (i.e., to develop green retrofitting) emerged. This indicates that when property management companies realize or analyze investment from a life-cycle perspective and regardless of the financial problem caused by high initial cost, the actual net value re-calculated benefit is sufficient for developers to promote green retrofitting. These results demonstrate that increased sufficient green retrofitting education and the promotion of the awareness of vulnerable green retrofitting are essential.

### 4.4. Discussion

The authors of this study proposed a new dynamic process by applying evolutionary game and determining a creative perspective for the research agents. This research applies occupant behavior to support more details of commercial buildings, such as contract disputes and cancellations between enterprises and property management groups. We found that the more detailed consideration of commercials indirectly affected the scale of the entire variables, the payoff matrix, and the ESS condition. Nonetheless, the government is the main impeller of green retrofitting, and the motivation of the government can be cancelled by excessive subsidy and welfare. In contrast, developers are only willing to take green retrofitting actions when there is sufficient economic benefit from the government, indicating a contradictory circumstance in balancing the benefit of agents. The benefit of the government and developers can be maximized by achieving a balanced stable situation. Additionally, the willingness of occupied enterprises was low regardless of the incentive level of external conditions.

In addition to the aforementioned points, this study presented several critical and comparative results compared to the findings of previous studies. Yang et al. [13] emphasized the importance of maximizing short-term benefits and increasing non-green retrofitting punishment. In contrast, this study revealed the importance of caring more and providing less punishment ratio to occupant enterprises owing to their vulnerable character. Similarly, Chen et al. [89] found that the government exhibits a limited positive behavior to green retrofitting and the feasibility to supply a more professional environment and increasing the awareness of green retrofitting to increase the perceived benefit, which can also be affected in the payoff matrix in the evolutionary game. Although incentives are predicted to be effective for residential buildings located in South Korea and have been confirmed to be synergistic between government and developer groups [11], there are several factors regarding commercial buildings, such as contract cancellation, which prevent developers from adopting a powerful retrofitting level. When the probability of multiple agents approaches one, the supportive behavior of all agents is impossible. Huang and Lin [66] reflect that the result of stable strategies is all based on economic consideration. This study reveals that the occupant groups exhibit less awareness and motivation, which is similar to the findings of a previous study on rural individuals in [66], further confirming our point that more friendly and economic benefits should be given to occupants.

Details can be seen in Table 8. According to the study of Kim et al. [11], carbon taxation is attracting attention for promoting green retrofitting. However, although developed economies, such as Norway, the UK, Sweden, and the UN, have established considerate and domestic market-based tax rates, taxation is still in the decision-making stage in China.

**Table 8.** Comparison of related studies (Note: G, D, O, B, and E correspond to government, developer, occupant bank, and ESCO enterprises, respectively.)

| Number | Year | Agent | Building Category | Region | Result and Recommendation | Ref. |
|---|---|---|---|---|---|---|
| 1 | 2019 | G/E | Residential | China | Unlimited synergistic strategy, a combination of positive and negative strategies, reducing costs | Yang et al. [13] |
| 2 | 2021 | G/E/O/B | Residential | China | Strategy defection threshold existing, undertaking of retrofitting risk, green financial atmosphere development, motivation towards the supply side | Chen et al. [89], Lu et al. [78] |
| 3 | 2022 | G/D | Residential | Korea | Unlimited synergistic strategy, more governmental budget, reducing costs, effect of increasing carbon tax | Kim et al. [11] |
| 4 | 2022 | G/E/O | Rural | China | Existence of strategy defection threshold, spark significance of the government, the main motivation of governmental subsidies | Huang and Lin [66] |
| 5 | 2022 | G/D/O | Commercial | China | Existence of strategy defection threshold, more beneficial occupant-friendly environment, more powerful popularity of the awareness of the potentials and benefits of green retrofit | This Research |

## 5. Conclusions and Recommendations

This research applied a trilateral government–PMC (developer) occupied enterprise relationship to demonstrate an evolutionary dynamic process. Simulation results obtained using MATLAB R2021b provided considerable enlightenment and inspiration to help enhance more effective green retrofit promotion. We employed a government–developer–occupant–agent–participant process using an evolutionary game model. In addition, we considered financial dispute problems associated with the formulation and cancellation of contracts between developers and occupants, which is one of the complexities of commercial buildings. More exoteric perspectives are presented below for future studies focusing on more detailed green retrofitting programs within varying building categories.

The policy presented in this study can be optimized using the following recommendations:

(1) The behavior between the government and the developer is inversely related to that between the developer and the occupant. Thus, an appropriate stimulation by the government can effectively incentivize developers to exhibit positive behavior. However, the government typically employs welfare distribution and punishment measures to push green retrofitting. We observed that these measures increase the financial cost to the government and increase pressure; thus, the government should provide a more relaxed national regulating policy environment to simultaneously balance the interaction relationship to maximize the benefit of the developer and mitigate pressure on the government. Particularly, the government is essential for implementing an appropriate stable condition to achieve the best-choice benefit.

(2) There was no difference in the attitude and behavior of occupants regardless of the negative or positive strength of the government and developers. In addition, under the normal market orientation condition, occupants gain limited benefit from a retrofitted building, but sustain a comparative high economic risk, including time-cost loss, as well as inevitable business operation impact due to the operating time of facility replacement in the renovation process. Without a preferential policy environment, occupants tend to choose a non-support behavior. Only a decrease in the incremental price difference between retrofitted buildings and traditional buildings and a reduction in the punishment ratio between developers and occupants could stimulate positive behavior in occupants and bring the evolutionary process to an ideal advanced direction.

(3) There are few studies on China's leading enterprises for implementing green building despite the existence of the model project. Although current studies have investigated technical specific energy-saving methods and economic method-based decision-making method, professional green industry practitioners are insufficient. In addition, the non-awareness of long-term LCC perspective investment idea may limit motivation because of the high initial investment cost. Within an amendatory annualized rate by the *NPV* method, developers are more willing to promote green retrofitting. An enlarged scale of the green AEC industry requires the establishment of more favorable policies by the government to revolutionize the AEC industry more thoroughly. Nevertheless, it is important to note that the high energy-saving potential is more significant than the leading model effect of green retrofitted buildings. Developers should strengthen green regeneration management in the entire life cycle of the project. A reasonable regeneration mode should be determined in the development stage to determine the technical advantages of the project to determine if the green investment target is essential. During the design stage, investment objectives should be implemented, and outdoor environment, monomer building, and details should be further established. Towards the construction stage, resources should be saved to a maximum extent to reduce pollution to ensure quality, progress, and cost. Furthermore, building functions should be maintained, operation energy consumption should be controlled, and market orientation development should be adjusted.

A limitation of this study is the lack of an actual case information to set the detailed parameters. The limited applicable retrofit economic data due to the privacy of contract

and operation conditions for developers and enterprises make it difficult to identify specific option differences of the changeset. The applicability and generalizability of commercial buildings with different contact and operation models and building characteristics differ and need to be specifically explored. Future studies may include testing and validating this parameter, and may focus on the verification of the social benefit value and the contact interaction value. Thus, occupied enterprises differ in scale, resulting in variation in the sensitivity levels of their comparative policies, which need to be further specialized.

**Author Contributions:** Conceptualization, S.-Y.W.; methodology, S.-Y.W.; software, S.-Y.W. and K.-T.L.; validation, S.-Y.W. and K.-T.L.; formal analysis, S.-Y.W.; investigation, S.-Y.W.; resources, S.-Y.W.; data curation, J.-H.K.; writing—original draft preparation, S.-Y.W.; writing—review and editing, K.-T.L. and J.-H.K.; visualization, S.-Y.W. and K.-T.L.; supervision, K.-T.L. and J.-H.K.; project administration, J.-H.K.; funding acquisition, J.-H.K. All authors have read and agreed to the published version of the manuscript.

**Funding:** This research was funded by Industrial consignment research (grant number 202100000001701).

**Institutional Review Board Statement:** Not applicable.

**Informed Consent Statement:** Not applicable.

**Data Availability Statement:** Data sharing not applicable. No new data were created or analyzed in this study. Data sharing is not applicable to this article.

**Conflicts of Interest:** The authors declare no conflict of interest.

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
