# Peer review of "Green Retrofitting Simulation for Sustainable Commercial Buildings in China Using a Proposed Multi-Agent Evolutionary Game"

_sustainability, doi:10.3390/su14137671_

Round 1

Reviewer 1 Report

Dear authors,

Thanks for your contribution to Sustainability.

Before further process of this manuscript, please check if it matches the scope of the journal.

With minor revisions of the manuscript, it might be accepted.

The opinions are set out below:

ENGLISH
The paper has several typos. Authors need to proofread the paper to eliminate all of them.
Some sentences are too long. Generally, it is better to write short sentences with one idea per sentence.

REFERENCES
The literature review is incomplete. Several relevant references are missing.

INTRODUCTION
The authors should add more references in the introduction to support the claims. Such as: Features of mobile apps for people with autism in a post covid-19 scenario: current status and recommendations for apps using AI, to investigate and enhance the literature. The authors need to better explain the context of this research, including why the research problem is important. Contributions should be highlighted more. It should be made clear what is novel and how it addresses the limitations of prior work.

RELATED WORK
The related work section is not well organized. Authors must try to categorize the papers and present them in a logical way. The authors should add a table that compares the key characteristics of prior work to highlight their differences and limitations. The authors may also consider adding a line in the table to describe the proposed solution.

 PROBLEM DEFINITION
The authors should add a clear and detailed problem definition. The authors should add an example to illustrate the problem definition.

METHOD
A novel solution is presented but it is important to better explain the design decisions (e.g., why the solution is designed like that). It is necessary to discuss the complexity of the proposed solution.

EXPERIMENTS 
The experiments should be updated to include some comparisons with newer studies.

Sincerely yours,

Author Response

Thank you for your comments. 

Reviewer 2 Report

This is an intriguing paper that explores green retrofitting simulation of commercial buildings. However, there are some points that can be improved.

- An extensive and pertinently literature review is carried out organized in different topics, but it must be placed before specifying the points on which the paper focuses. In this way, the authors can clarify more in detail what are the specific contributions of the research to the previous knowledge, and thus justify more in deep the objectives and methodology.

- In relation to the simulation model developed by the authors, it would be appropriate to also specify more clearly the advances and differences with respect to previous models, in order to clarify the relevance and degree of innovation of the research’s contributions.

- Although the relationship with real behavior of the simulation model is not sufficiently contrasted, I think it is appropriate the conclusions have pointed out the limitations and future research developments.

Author Response

Thank you for your comments. 

Round 2

Reviewer 2 Report

The paper has been sufficiently improved to meet the requirements of a scientific journal